# Ca²⁺ inactivation of the mammalian ryanodine receptor type 1 in a lipidic environment revealed by cryo-EM

**Ashok R Nayak, Montserrat Samsó***

Department of Physiology and Biophysics, Virginia Commonwealth University, Richmond, United States

**Abstract** Activation of the intracellular Ca²⁺ channel ryanodine receptor (RyR) triggers a cytosolic Ca²⁺ surge, while elevated cytosolic Ca²⁺ inhibits the channel in a negative feedback mechanism. Cryogenic electron microscopy of rabbit RyR1 embedded in nanodiscs under partially inactivating Ca²⁺ conditions revealed an open and a closed-inactivated conformation. Ca²⁺ binding to the high-affinity site engages the central and C-terminal domains into a block, which pries the S6 four-helix bundle open. Further rotation of this block pushes S6 toward the central axis, closing (inactivating) the channel. Main characteristics of the Ca²⁺-inactivated conformation are downward conformation of the cytoplasmic assembly and tightly knit subunit interface contributed by a fully occupied Ca²⁺ activation site, two inter-subunit resolved lipids, and two salt bridges between the EF hand domain and the S2–S3 loop validated by disease-causing mutations. The structural insight illustrates the prior Ca²⁺ activation prerequisite for Ca²⁺ inactivation and provides for a seamless transition from inactivated to closed conformations.

## Editor's evaluation

This study provides insights into structural changes leading to calcium-dependent inactivation (CDI) in type 1 ryanodine receptors (ryR1). The results nicely rationalize how some disease-causing mutations in RyR1 eliminate CDI of the channel and will be of interest to ion channel structural biologists and physiologists studying skeletal muscle pathologies.

*For correspondence:
montserrat.samso@vcuhealth.
org

**Competing interest:** The authors declare that no competing interests exist.

## Introduction

Ryanodine receptors (RyRs) are complex intracellular, multidomain Ca²⁺ channels that generate Ca²⁺ transients in cells of higher metazoans. They are pivotal for the contraction of skeletal and cardiac muscles (*Flucher and Franzini-Armstrong, 1996*; *Meissner, 2017*; *Ríos, 2018*), supporting a transient rise in cytosolic Ca²⁺ from its resting level of ~100 nM, which is driven by a more than 1000-fold Ca²⁺ concentration gradient across the membrane of the endoplasmic reticulum (sarcoplasmic reticulum in muscle [SR]) (*Ziman et al., 2010*). RyRs also play a role in neuron excitability (*Albrecht et al., 2001*; *Arias-Cavieres et al., 2018*; *Bouchard et al., 2003*) and a myriad of other Ca²⁺-dependent pathways such as differentiation, survival, and apoptosis (*Bagur and Hajnóczky, 2017*; *Tu et al., 2016*). Dysregulation of the channel leads to several life-threatening diseases such as malignant hyperthermia, central core disease, sudden cardiac death, and lethal fetal akinesia deformation sequence syndrome (*Alkhunaizi et al., 2019*; *McCarthy et al., 2000*; *Priori et al., 2001*; *Tiso et al., 2001*; *Treves et al., 2008*).

The high-conductance RyR channel is strongly regulated, with cytosolic Ca²⁺ concentration having a biphasic effect on the probability of channel opening. Notably, both intracellular Ca²⁺ channels, RyRs and inositol P₃ receptors (IP₃Rs), display a bell-shaped curve of Ca²⁺ dependence (*Bezprozvanny*

*et al., 1991*; *Meissner, 2017*; *Meissner et al., 1986*), characteristic of dual regulation by cytosolic $Ca^{2+}$. In RyR, the ascendant branch of $Ca^{2+}$ dependence with half-maximal concentration ($K_a$) of 1–5 µM (*Laver, 2018*; *Laver and Lamb, 1998*) is mediated by a high-affinity cytosolic $Ca^{2+}$ site (*des Georges et al., 2016*). $Ca^{2+}$ concentrations above 100 µM, reached in the tight nanodomain surrounding the RyR upon its opening (*Langer and Peskoff, 1996*), inhibit the channel (*Chen et al., 1997*; *Xu and Meissner, 1998*; *Yamaguchi, 2020*), which rapidly enters a refractory period that has been proposed to prevent depletion of the SR $Ca^{2+}$(*Bers, 2002*; *Ríos et al., 2008*). Out of the three isoforms, the 'skeletal muscle' RyR1 isoform studied here is the most sensitive to $Ca^{2+}$ inactivation (*Meissner, 2017*), where inactivation-impairing mutations are known to cause malignant hyperthermia (*Gomez et al., 2016*). While the inactivation of RyR by $Ca^{2+}$ has been characterized at the functional level, its underlying mechanism is unknown. Outstanding questions are whether high $Ca^{2+}$ induces an allosteric change in RyR that dampens the affinity of its activation site for $Ca^{2+}$ and whether the $Ca^{2+}$-inactivated conformation is similar or distinct with respect to the closed conformation obtained in the absence of $Ca^{2+}$.

Here, cryogenic electron microscopy (cryo-EM) and single-particle 3D reconstruction were applied to rabbit RyR1 embedded in nanodiscs under conditions of partial $Ca^{2+}$ inactivation. Classification of two independent cryo-EM datasets revealed, in both cases, the coexistence of closed and open conformations, in agreement with functional experiments performed on the same channels using tritiated ryanodine binding. In the closed (inactivated) conformation, the resolution of the cryo-EM maps of nanodisc-embedded RyR1 enabled visualization of two lipids buried in a pocket of the transmembrane domain (TMD). To our knowledge, this is the first time that lipid is visualized in direct contact with the RyR. The open state, obtained by classification of the same dataset, represents the first RyR1 open conformation achieved in the absence of any extra activator other than the physiological activators, $Ca^{2+}$ and ATP. We also carried out a control 3D reconstruction of RyR1 under the same conditions except for the absence of $Ca^{2+}$, which yielded a closed channel. A comparison of the closed conformations of RyR1 at high $Ca^{2+}$ and of RyR1 in the absence of $Ca^{2+}$ revealed unique features associated to $Ca^{2+}$-inactivation. Both $Ca^{2+}$-induced activation and inactivation can be explained by a unifying mechanism that involves conformational rearrangements within the central region of RyR1. Thus, the 3D structures of closed, open, and inactivated RyR1 embedded in lipidic nanodisc provide a mechanistic framework to understand the biphasic response of RyR1 to $Ca^{2+}$. In addition, two inter-subunit salt bridges appear to mediate the $Ca^{2+}$-inactivated structural rearrangement, a finding supported by disease-causing mutations hindering such interactions and known to impair $Ca^{2+}$-induced inactivation of the RyR1.

## Results
### Experimental design and functional validation of RyR1

Experimental conditions were fine-tuned to obtain the inactivated state. The channels were prepared in 2 mM free $Ca^{2+}$, a concentration that inactivates the channel. As RyR1 is constitutively bound to the sensitizing ATP in the muscle cell (*Kushmerick et al., 1992*; *Meissner et al., 1986*; *Xu et al., 1996*), we included its non-hydrolyzable form, AMP-PCP (ACP), for its structural determination. In order to visualize the direct effect of $Ca^{2+}$ on RyR1's conformation, FKBP12, a stabilizer of the cytoplasmic domain, was excluded. To determine the structure of RyR1 in a detergent-free, membrane-embedded natural state, RyR1 was reconstituted into membrane scaffold protein (MSP) 1E3D1 nanodiscs in the presence of phosphatidylcholine, an abundant phospholipid in membrane fraction preparations. As a control, we carried out cryo-EM and 3D reconstruction of RyR1 using identical buffer conditions and saturating ACP, and substituted $Ca^{2+}$ by 1 mM EGTA plus 1 mM EDTA (dataset denominated RyR1-ACP/EGTA). The channels were also reconstituted into nanodiscs in the presence of phosphatidylcholine.

The $Ca^{2+}$-induced activity profile of rabbit RyR1 in SR membranes was determined using the tritiated ryanodine binding assay, which reflects the probability of channel opening. The assay indicated RyR1 channel inactivation above 0.1 mM $Ca^{2+}$, with an $IC_{50}$ of 0.6 mM (*Figure 1A*). The presence of 2 mM ATP increased the efficacy of $Ca^{2+}$-induced activation by approximately threefold at peak activation with similar potency for $Ca^{2+}$-induced inactivation ($IC_{50}$ of 0.7 mM), in agreement with earlier results obtained in lipid bilayer (*Laver et al., 1995*). We replicated the ryanodine-binding experiments over the same $Ca^{2+}$ concentration range with 2 mM ACP. Maximal ryanodine binding was ~1.5-fold

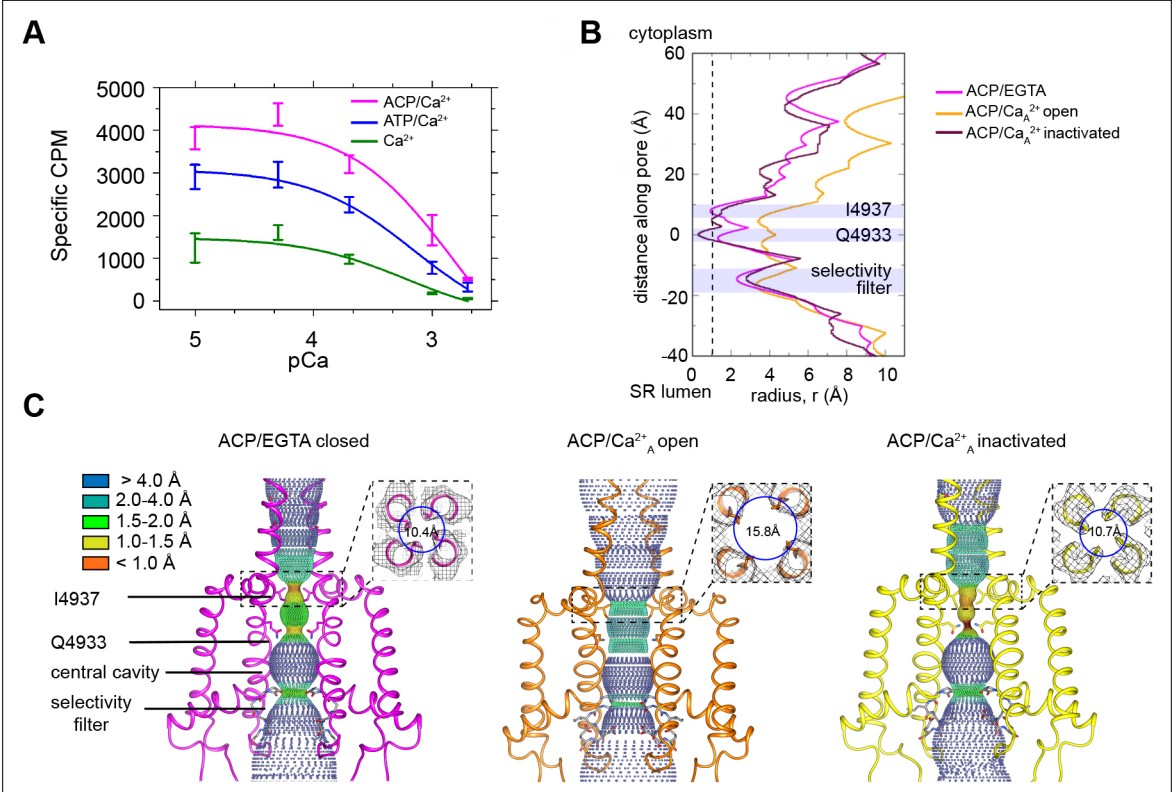

**Figure 1.** RyR1 at high Ca$^{2+}$ concentration exhibits a mixture of open and inactivated conformations. (**A**) Ryanodine binding of rabbit skeletal sarcoplasmic reticulum (SR) microsomes showing the Ca$^{2+}$-induced inactivation of RyR1; activators ATP or ACP (2 mM) increased channel open probability by 3- and 4.5-fold, respectively. Ca$^{2+}$ concentrations of 100 μM to 2 mM progressively decrease the probability of opening (P$_o$). Mean specific [$^3$H]-ryanodine binding ± SEM from four independent experiments. (**B**) Pore profile of RyR1-ACP/EGTA, and of open and inactivated conformations in the RyR1-ACP/Ca$^{2+}_A$ dataset calculated with the program HOLE (*Smart et al., 1993*). The position of relevant landmarks is indicated with violet shadowing. Radius corresponding to a dehydrated Ca$^{2+}$ ion shown with a dashed line. (**C**) Dotted surfaces of RyR1 ion permeation pathway in closed, open, and inactivated conformations, color-coded according to pore radius. Overlaid coordinates of the S5, S6, pore helices, and selectivity filter are shown for two diagonal protomers. Insets: cytoplasmic views of the Ile4937 constriction and corresponding pore diameter measured at the Cα backbone. Similar results were obtained for RyR1-ACP/Ca$^{2+}_B$ (*Figure 1—figure supplement 5*).

The online version of this article includes the following figure supplement(s) for figure 1:

**Figure supplement 1.** Image processing scheme for the RyR1-ACP/Ca$^{2+}_A$ dataset.

**Figure supplement 2.** Correlation of RyR1-ACP/Ca$^{2+}_A$ inactivated model with the cryogenic electron microscopy (cryo-EM) density.

**Figure supplement 3.** Image processing scheme for the RyR1-ACP/Ca$^{2+}_B$ dataset.

**Figure supplement 4.** Image processing scheme for the RyR1-ACP/EGTA dataset.

**Figure supplement 5.** Pore conformation in the RyR1-ACP/Ca$^{2+}_B$ dataset.

higher in the presence of ACP compared to ATP, and in this case slightly higher Ca$^{2+}$ concentration was necessary for the same degree of RyR1 inhibition, with an IC$_{50}$ value for Ca$^{2+}$ of 1.5 mM in the presence of ACP. Thus, under our experimental conditions of 2 mM free Ca$^{2+}$, RyR1 inactivation was partial.

## Classification of the high Ca$^{2+}$ dataset reveals an *open* conformation and a *closed-inactivated* conformation

Cryo-EM of rabbit RyR1 embedded in nanodisc in the presence of 2 mM ACP and 2 mM free Ca$^{2+}$ followed by single-particle analysis and 3D classification revealed two classes of particles, with their pore either fully open or fully closed. The findings, reproduced in two independent datasets (A and B), reflected the partial inhibition observed at 2 mM free Ca$^{2+}$ (pCa 2.7) in the ryanodine-binding studies (*Figure 1A*). The 3D reconstructions of RyR1-ACP/Ca$^{2+}$ open reported here are the first RyR1 open structures obtained in the presence of natural activators alone, possibly owing to the use of nanodiscs.

The RyR1-ACP/Ca$^{2+}$ reconstructions with a closed pore suggest a distinct conformation that we refer to as RyR1-ACP/Ca$^{2+}$ inactivated henceforth, as further supported by our data.

A symmetry of C4 was imposed after ascertaining the true fourfold symmetry of the protein. The closed-pore (inactivated) channels were resolved to 3.8 Å and 4.1 Å resolution for the A and B datasets, respectively, which improved to 3.5 Å and 3.8 Å, respectively, using a phase improvement procedure (*Terwilliger et al., 2020*). Despite this improvement, we followed a conservative approach and used the non-density-modified maps for subsequent analysis. The open conformation represented a smaller fraction of the data (13 and 19% for the A and B datasets), which limited the resolution to 4.6 Å and 5.8 Å, respectively (*Figure 1—figure supplement 1*, *Figure 1—figure supplement 2*, *Figure 1—figure supplement 3*). The RyR1-ACP/Ca$^{2+}$$_A$ open subset improved to a resolution of 4.0 Å after a symmetry expansion step. Unless specified, the reported observations correspond to dataset A.

Classification of the control RyR1-ACP/EGTA dataset yielded a single class with a resolution of 4.3 Å (0.143 FSC gold standard), but further symmetry expansion and focused classification using a quadrant-shaped mask increased resolution to 3.9 Å (*Figure 1—figure supplement 4*). Despite the presence of the ACP activator, the 3D reconstruction of RyR1-ACP/EGTA yielded a closed state. *Tables 1 and 2* summarize the cryo-EM data collection, single-particle image processing, model quality attributes, and database IDs for all datasets, and *Table 3* compiles the main characteristics of the three conformations analyzed (closed, open, inactivated).

Pore analysis using the program HOLE (*Figure 1B and C*) of RyR1-ACP/EGTA indicated a closed pore with a radius constricting to 1 Å at the known hydrophobic gate Ile4937. In both RyR1-ACP/Ca$^{2+}$ open A and B datasets, the hydrophobic gate as appraised by the pore profile (*Figure 1B*, *Figure 1—figure supplement 5*) and as measured at the S6 helix backbone C$_\alpha$ atoms (for more accurate measurement according to the lower resolution of the two datasets) (*Figure 1C*, *Figure 1—figure supplement 5*) had diameters of 15.8 Å and 16.0 Å, which indicates an open, Ca$^{2+}$-permeable pore in both cases. Pore dimensions are comparable to the pore diameter of RyR1 open structures obtained in the presence of activators such as Ca$^{2+}$/PCB95 or Ca$^{2+}$/caffeine/ATP (16.7 Å; PDB ID: 5TAL; *des Georges et al., 2016*). The slightly narrower pore in our case could be attributed to the millimolar instead of submicromolar Ca$^{2+}$ in our case and/or extra activators in addition to endogenous physiological activators in earlier open structures.

The reconstructions corresponding to RyR1-ACP/Ca$^{2+}$ inactivated of the A and B datasets displayed a pore diameter at the Ile4937 gate of ~2 Å when considering the side chains, and 10.7 Å and 11.3 Å diameter, respectively, when measured at the backbone C$_\alpha$ atoms, making it impermeable to Ca$^{2+}$ ions (*Figure 1B and C*, *Figure 1—figure supplement 5*). The position of the S6 backbone at Ile4937 is similar to our closed structure ($r$ = 10.4 Å) and to RyR1-FKBP12/EGTA ($r$ = 10.3 Å; PDB ID: 5TB0; *des Georges et al., 2016*). Interestingly, the channel pore was narrower than 1 Å at Gln4933 in RyR1-ACP/Ca$^{2+}$$_A$ inactivated (*Figure 1B and C*).

We previously established that the large square-shaped cytoplasmic shell of the RyR undergoes a conformational change upon opening. The periphery of its four quadrants tilts downward (toward the membrane), while its inner corner tilts away from it, rotating around a pivot point (*Samsó et al., 2009*). Such tilt can be quantified with the flexion angle, whereby negative angle (downward) correlates with opening, positive angle with closing, and absence of FKBP lowers the flexion angle of closed states (*Steele and Samsó, 2019*). In general, the approximate ranges of flexion angles are +1° to +2° for RyR1-FKBP12/EGTA closed, –1° to –3° for RyR1-EGTA closed, 0° to –3° for RyR1-FKBP12 'primed' (with activating ligands and closed pore), and –1.5° to –5° for RyR1-Ca$^{2+}$ open with or without FKBP12 (*Iyer et al., 2020*; *Steele and Samsó, 2019*). Here, RyR1-ACP/Ca$^{2+}$ open channels have flexion angles of –5.1° and –4.8° for the A and B datasets, within the expected range. The consensus 3D reconstruction of RyR1-ACP/EGTA had flexion angle of –2.2°, and its two classes had flexion angles of - 1.8° (24%) and –2.9° (76%) (*Figure 1—figure supplement 4*). But unexpectedly for closed-pore channels, RyR1-ACP/Ca$^{2+}$ inactivated datasets have a negative flexion angle (–4.6° and –4.2° for A and B datasets, respectively) (*Figure 1—figure supplements 1 and 3*). As there was an indication of variability in the cytoplasmic shell of RyR1-ACP/Ca$^{2+}$$_A$-inactivated dataset, we carried out fourfold symmetry expansion and focused classification for the monomer and obtained three subclasses, all showing downward motion: class 1 (22% of particles, 3.7 Å resolution, –3.6° flexion angle), class 2 (44% of particles, 3.3 Å resolution, –4.4° flexion angle), and class 3 (20% of particles, 3.8 Å resolution,

**Table 1.** Summary of cryogenic electron microscopy (cryo-EM) data collection and image processing parameters.

| Dataset | RyR1 ACP/EGTA closed | RyR1 ACP/Ca2+ A inactivated | RyR1 ACP/Ca2+ A inactivated class1 | RyR1 ACP/Ca2+ A inactivated class2 | RyR1 ACP/Ca2+ A inactivated class3 | RyR1 ACP/Ca2+ A inactivated CDTM | RyR1 ACP/Ca2+ A open | RyR1 ACP/Ca2+ A open CDTM | RyR1 ACP/Ca2+ B inactivated | RyR1 ACP/Ca2+ B open |
|---|---|---|---|---|---|---|---|---|---|---|
| **Data acquisition** | | | | | | | | | | |
| Microscope/detector | Krios/K2 | Krios/K3 | | | | | | | Krios/K2 | |
| Voltage (kV) | 300 | 300 | | | | | | | 300 | |
| Magnification | 130,000 | 81,000 | | | | | | | 130,000 | |
| Defocus range (μm) | −1.2 to −2.2 | −1.25 to −2.5 | | | | | | | −1.25 to −2.5 | |
| Pixel size (Å) (calibrated) | 1.06 (1.07) | 1.08 (1.105) | | | | | | | 1.06 (1.07) | |
| Total electron dose (e/Å²) | 70 | 70 | | | | | | | 70 | |
| Exposure time (s) | 12 | 4.4 | | | | | | | 14 | |
| Number of frames | 60 | 50 | | | | | | | 50 | |
| Total number of Micrographs | 1959 | 10,002 | | | | | | | 1346 | |
| **Image processing** | | | | | | | | | | |
| Total number of particles selected | 52,856 | 311,258 | | | | | | | 35,554 | |
| Final number of particles | 21,551 | 90,530 | 79,277 | 159,444 | 71,499 | 90,530 | 14,012 | 14,012 | 14,669 | 4160 |
| Reconstruction symmetry | C4 | C4 | C1 | C1 | C1 | C4 | C4 | C4 | C4 | C4 |
| Map resolution, FSC (0.143) (symmetry expanded map) | 4.3 (3.9) | 3.8 (3.5) | 3.7 | 3.3 | 3.8 | 3.5 | 4.6 (4.0) | 4.4 | 4.1 (3.8) | 5.8 |
| Map sharpening B-factor (Å²) (symmetry expanded map) | −132 | −150 (−144) | −125 | −102 | −123 | −142 | −168 (−135) | −144 | −111 (−89) | −148 |
| EMDB ID | 22616 22597 | 25828 | 25831 | 25830 | 25832 | 25828* | 25829 | 25829* | 25833 | - |

*Submitted as an additional map.

**Table 2.** Summary of model refinement and validation statistics.

| Dataset | RyR1 ACP-EGTA | RyR1 ACP/Ca2+A inactivated | RyR1 ACP/Ca2+ A inactivated class1 | RyR1 ACP/Ca2+ A inactivated class2 | RyR1 ACP/Ca2+ A inactivated class3 | RyR1 ACP/Ca2+ A open |
|---|---|---|---|---|---|---|
| RMS deviation (bonds) | 0.004 | 0.086 | 0.004 | 0.004 | 0.003 | 0.004 |
| RMS deviation (angle) | 0.919 | 0.812 | 0.893 | 0.788 | 0.848 | 0.766 |
| Ramachandran plot statistics (%) | | | | | | |
| Preferred | 92.39 | 93.38 | 93.85 | 95.54 | 94.13 | 94.90 |
| Allowed | 7.45 | 6.57 | 5.97 | 4.38 | 5.75 | 5.07 |
| Outliers | 0.16 | 0.05 | 0.19 | 0.08 | 0.12 | 0.03 |
| Clash-score | 7.58 | 7.65 | 5.62 | 3.90 | 4.94 | 4.60 |
| MolProbity score | 1.89 | 1.85 | 1.71 | 1.49 | 1.65 | 1.59 |
| PDB ID | 7K0T | 7TDG | 7TDJ | 7TDI | 7TDK | 7TDH |

–5.6° flexion angle) (*Figure 1—figure supplement 1*). Such pronounced downward flexion angles in a closed channel cannot be explained just by the lack of FKBP12 in our preparations. The distinct conformation of RyR1-ACP/Ca$^{2+}$ inactivated, consisting of a closed pore and extreme-downward cytoplasmic assembly, prompted further analysis of the central region that joins the cytoplasmic and transmembrane domains.

## Different arrangement of the central region in Ca$^{2+}$-inactivated, closed and open conformations

In RyR1-ACP/Ca$^{2+}$ open, the high-affinity Ca$^{2+}$-binding site was formed by Glu3967, Glu3893 (from the CD; residues 3668–4070), and Thr5001 (from the CTD (C-terminal domain); residues 4957–5037) as reported earlier (*des Georges et al., 2016*), and additionally Gln3970 in our case, which was visible up to 20σ. The Ca$^{2+}$-induced reorientation of CTD with respect to CD and subsequent separation of S6 was obvious when compared to the RyR1-ACP/EGTA structure (*Figure 2A*, *Figure 2—figure supplement 1*), consistent with previous reports of Ca$^{2+}$-induced activation (*Bai et al., 2016*; *des Georges et al., 2016*).

**Table 3.** Differences between closed, open, and inactivated conformations.

| Conformation | Closed | Open | Inactivated |
|---|---|---|---|
| Pore | Closed | Open | Closed |
| Flexion angle (average) | –2.2° | –5.3° | –4.4° |
| In-plane rotation | 0 (reference) | Counterclockwise | Counterclockwise |
| High affinity Ca$^{2+}$ 900 site | Empty | Occupied | Occupied |
| Ca$^{2+}$ bound to ACP | No | Yes | Yes |
| CD/CTD | Disconnected | Connected | Connected |
| Position of CD-C' | Down | Up/intermediate | Up |
| Position of CD-C' | Up | Down/intermediate | Down |
| EF hand/S2–S3 loop | Separated | Separated | Form two salt bridges |
| S6 TMD helix (4921-4928) | π | π wide-short | π narrow-long |
| Lipids in TMD crevice | Present | Absent | Present |

TMD: transmembrane domain.

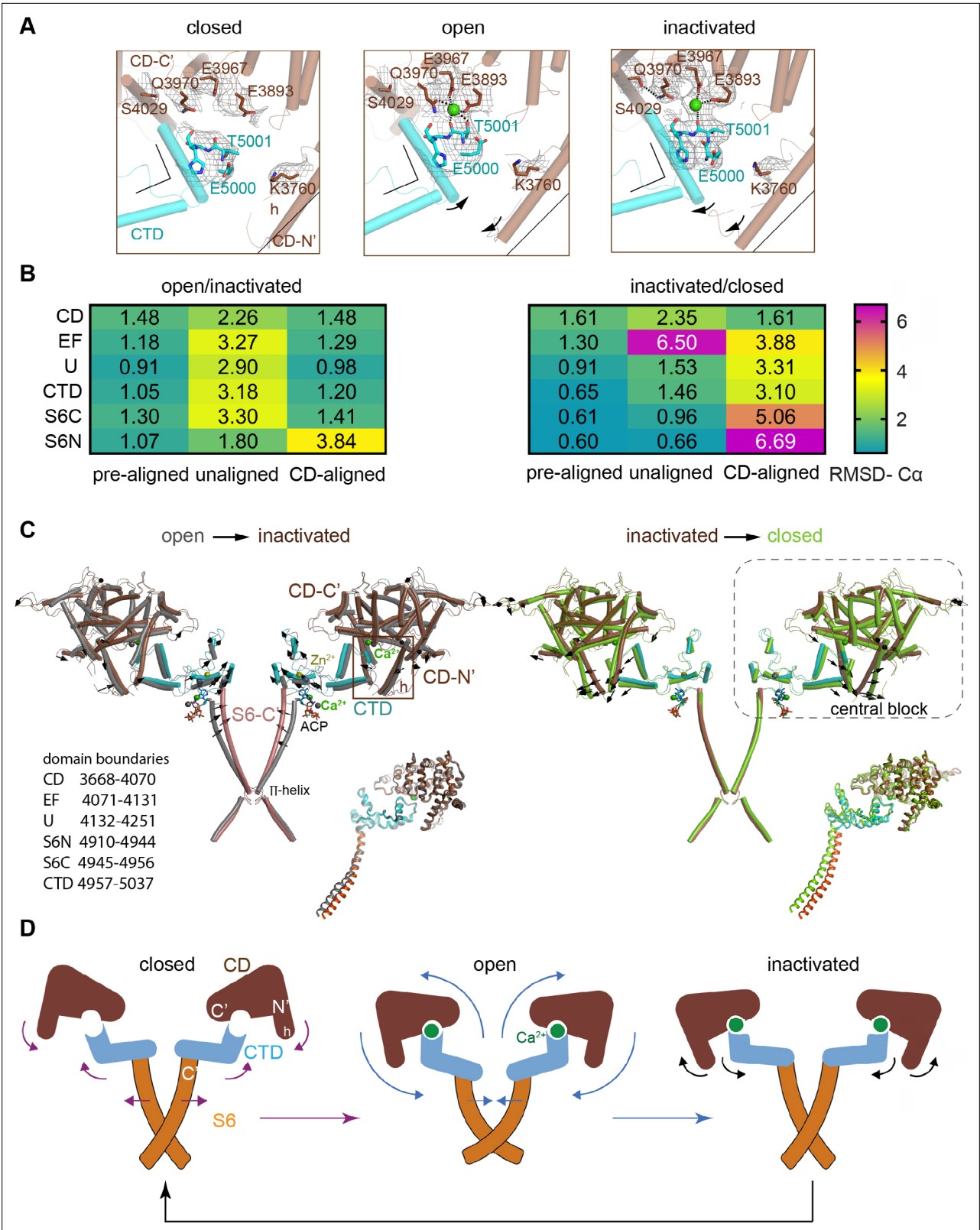

**Figure 2.** Inactivation of RyR1 involves out-of-plane rotation of the central block and rearrangement around the $Ca^{2+}$ activation site. (**A**) The high-affinity $Ca^{2+}$-binding site in the CD/CTD interface with density around the $Ca^{2+}$ site contoured at 4σ. Contacts within 2.8 Å from $Ca^{2+}$, as well as additional contact Gln3970-Ser4029 within 3.6 Å, are represented by dashed lines. Channel axis is on the left. During the transition from open to inactivated conformations, the CD/CTD block tilts around the $Ca^{2+}$-binding site such that the protruding fourth helix of the CD (h) and connected CTD tilt inward,

*Figure 2 continued on next page*

*Figure 2 continued*

while the CD-C′ tilts upward and away from the sarcoplasmic reticulum (SR) membrane – with Gln3970 separating from Ca²⁺ by ~6 Å. Arrows and stationary reference lines illustrate the conformational changes undergone with respect to the panel on the left. The region represented relative to the channel is highlighted with a square in panel (**C**). See *Figure 2—figure supplement 1* for the corresponding space-filling representation. (**B**) Heat map showing Cα-backbone root mean square deviation (RMSD) (in Å) between domain pairs from respective conformations after aligning them (pre-aligned), in their native conformation (unaligned), and after aligning the protomers through their respective CD (CD-aligned). The RMSD difference between unaligned and pre-aligned represents the change caused by domain relocation. (**C**) Overlaid structures of the CD-CTD-S6 domains in different conformations; only two protomers shown for clarity. Left: in the transition from RyR1-ACP/Ca²⁺ open (gray) to RyR1-ACP/Ca²⁺ inactivated (colored), the CD-CTD block, 'connected' by Ca²⁺ coordination, undergoes an out-of-plane rotation around the Ca²⁺-binding site that pushes S6C′ toward the pore axis, closing the channel. Right: in the transition from RyR1-ACP/Ca²⁺ inactivated (colored) to RyR1-ACP/EGTA (green), Ca²⁺ unbinding disconnects the CD from the CTD. Structures at the bottom right of each panel show the comparison of the CD-CTD-S6C′ of the central block after forcing superimposition of their respective CDs. Residue boundaries for relevant domains are specified. (**D**) Schematics of the conformational changes from RyR1-ACP/EGTA (closed), to RyR1-ACP/Ca²⁺ open, to RyR1-ACP/Ca²⁺ inactivated. Activation, where Ca²⁺ binds to the high-affinity site, is required prior to inactivation. The CD-protruding fourth helix (3753–3769) is indicated by 'h.' Colored arrows indicate the conformational change toward the following structure.

The online version of this article includes the following video and figure supplement(s) for figure 2:

**Figure supplement 1.** Reorganization of the high-affinity Ca²⁺-binding site at the CD/CTD interface under different conditions.

**Figure supplement 2.** Rotation axes of the CD of RyR1 among the different conformational transitions.

**Figure 2—video 1.** Ca²⁺-induced transitions in activation and inactivation.

https://elifesciences.org/articles/75568/figures#fig2video1

An outstanding question is how higher Ca²⁺ concentration may result in inactivation. Potential mechanisms are an additional allosteric change that dampens the affinity of the Ca²⁺ activation site, or that the high-affinity Ca²⁺ site remains occupied while an additional conformational change overcomes activation. Consistent with the second scenario, the high-affinity Ca²⁺-binding site in RyR1-ACP/Ca²⁺ inactivated remained fully occupied by Ca²⁺ up to 20σ, although with additional out-of-plane tilting of the CD and CTD around the Ca²⁺-binding site. The Ca²⁺ ion was coordinated by Glu3893, Glu3967 (CD), and Thr5001 (CTD) within contact distances of 2.8 Å, consistent with the open structures. However, on the CD-C′ side, Gln3970 lost coordination to Ca²⁺, interacting with Ser4029 instead (*Figure 2A*, *Figure 2—figure supplement 1*). Mutations Q3970E/K in RyR1 are implicated in central core disease and equivalent RyR2 mutations in cardiac arrhythmia (*Chirasani et al., 2019*), which highlights the important Ca²⁺ sensing role for this residue.

To understand how the fully occupied high-affinity Ca²⁺-binding site (CD/CTD interface) led to a closed pore in RyR1-ACP/Ca²⁺ inactivated, we looked for differences between RyR1-ACP/Ca²⁺ open and RyR1-ACP/Ca²⁺ inactivated in the domains spanning from the CD/CTD interface to the pore. Specifically, we focused on domains around the high-affinity Ca²⁺-binding site (CD, EF hand, U-motif, CTD, and the C′ section of S6 or S6C′; domain boundaries indicated in *Figure 2*), which we term 'central block.' A comparison was done for dataset A after confirming similarity for the central block of 0.95 Å Cα root mean square deviation (RMSD) for the encompassed 672 resolved residues, between inactivated A and B datasets (*Figure 2—figure supplement 2A*). The RMSD (Cα backbone) between individual domains from RyR1-ACP/Ca²⁺ inactivated versus open was below 1.5 Å for all comparisons when the domains were pre-aligned pairwise (*Figure 2B*, *Figure 2—figure supplement 2B and C*), indicating little conformational change. The N′ section of S6 was also included in the comparisons. Without pre-alignment, RMSD was higher (*Figure 2B*), indicating domain repositioning, the degree of which was estimated by subtracting the pre-aligned from the unaligned RMSDs. This yielded an average shift of ~2 Å per domain in going from RyR1-ACP/Ca²⁺ open to RyR1-ACP/Ca²⁺ inactivated, except for S6N′ where the shift was ~0.7 Å (*Figure 2B*). Comparing the same domains between RyR1-ACP/Ca²⁺ inactivated and RyR1-ACP/EGTA (closed) yielded smaller repositioning with average shifts of ~1 Å, reflecting similarity in the domain's locations, with the exception of the EF hand domain that undergoes an ~5 Å repositioning (*Figure 2B*, *Figure 2—figure supplement 2C*).

When the central blocks were pre-aligned as a unit using their respective CD domains, and the individual domains were compared pairwise (CD-aligned; *Figure 2—figure supplement 2A*), there was a good overlap between RyR1-ACP/Ca²⁺ open and RyR1-ACP/Ca²⁺ inactivated (RMSD below 1.5 Å) except for S6N′ (3.8 Å), revealing the bend in the middle of S6 upon opening. In contrast, the CD-aligned RMSD of RyR1-ACP/Ca²⁺ inactivated vs. RyR1-ACP/EGTA closed was higher, between 3

and 4 Å for the EF hands, U-motif, and CTD, and increases toward the C-terminus, 5 Å for S6C' and almost 7 Å for S6N' (*Figure 2B*, 'CD-aligned'). This indicated that domains CD, EF hands, U-motif, S6C', and CTD relocated together, and that the central block became more compact in the presence of $Ca^{2+}$ acting at the CD/CTD interface. Taken together, these results imply that conformational changes from CD are transmitted to S6 only in the presence of $Ca^{2+}$ (*Figure 2C and D*).

We analyzed how the central block may evolve from RyR1-ACP/EGTA (closed), to RyR1-ACP/$Ca^{2+}$ open, to RyR1-ACP/$Ca^{2+}$ inactivated. Upon $Ca^{2+}$-induced opening, 'engagement' of CD and CTD by $Ca^{2+}$ tilts the CD out of plane (such that helix h tilts 3° inwards), while the CTD tilts up and outwards. This pulls the S6 helices (which are directly connected to the CTD) away from the pore axis, opening the ion pathway (*Figure 2C and D*). To reach the inactivated state, the central block tilts further (see rotation axis in *Figure 2—figure supplement 2B*) in a movement similar to pushing down the levers of a winged corkscrew, which pushes each S6 helix 2.5 Å toward the pore axis, closing the channel. Alpha helices within the CTD act as a lever coupling this out-of-plane rotation to S6C' and as described further below, ATP reinforces the connection between CTD-S6C' in the presence of $Ca^{2+}$. These conformational changes involved in activation, inactivation, and transition from the inactivated to the closed state are shown in the morph among conformations in *Figure 2—video 1* and illustrated in the schematics in *Figure 2D*.

## Formation of salt bridges between EF hand domain and S2–S3 loop of the neighboring subunit in the inactivated state

The EF hand domain of RyR1, a candidate $Ca^{2+}$ regulation site (*Du et al., 1998*; *Gomez et al., 2016*; *Gomez and Yamaguchi, 2014*; *Xiong et al., 1998*), did not show significant changes, with maximum Cα pre-aligned RMSD of 1.5 Å among the different states analyzed (closed, open, inactivated). The EF hand loops (residues 4081–4090, 4116–4123) were empty in our high $Ca^{2+}$ conditions, which is consistent with the low affinity of $Ca^{2+}$ to this site ($K_d$ 3.7 mM, *Xiong et al., 1998*). Nonetheless, the entire EF hand domain, which protrudes from the CD-C', repositions noticeably during activation, with a 3.4° counterclockwise in-plane rotation (as seen from the cytoplasmic side), and 6.8° upward out-of-plane rotation. This movement brings the EF hand domain in closer proximity to the S2–S3 loop (cytoplasmic loop between S2 and S3 TMD helices; residues 4664–4786) (*Figure 3*). With inactivation, an additional 1° counterclockwise rotation and 2.7° upward out-of-plane rotations define a physical limit to the counterclockwise motion, forming two salt bridges at this inter-subunit interface: Glu4075-Arg4736 and Lys4101-Asp4730, with their side chains within 3.5 Å (*Figure 3A*). This interaction is critical to support inactivation, as demonstrated by the fact that MH/CCD mutations facing this interface F4732D, G4733E, and R4736W/Q, with the latter including the Arg directly involved in the salt bridge (*Figure 3B*), greatly reduced channel inactivation (*Gomez et al., 2016*). On the other hand, MH/CCD mutations T4082M, S4113L, and N4120Y, in regions of the EF hand domain away from the interface (*Figure 3B*), did not affect RyR1 inactivation (*Gomez et al., 2016*), serving as a negative control for this hypothesis.

## $Ca^{2+}$ binds to the ATP-binding pocket

The ATP-binding site showed full occupancy in the three conformations – open, closed, and inactivated. Under closed-state conditions, ACP bound to the pocket formed by the U-motif, S6C', and CTD (*Figure 4A*, 'closed'), in the same position as ATP (*des Georges et al., 2016*), with a high map significance (7σ). An additional elongated non-protein density was associated to ACP in the RyR1-ACP/$Ca^{2+}$ open and inactivated maps. This distinct density has high map significance (15σ) (*Figure 4A*, *Figure 4—figure supplement 1*), suggestive of a putative $Ca^{2+}$ ion. Well-resolved local density in RyR1-ACP/$Ca^{2+}{}_A$-inactivated class 1 map allowed tentative modeling of two waters on either side of the putative $Ca^{2+}$ that connect on either side to ACP's γ-phosphate and Thr4979 (CTD) (*Figure 4A*, 'inactivated'), suggesting potential coordination through $Ca^{2+}$'s first layer of hydration. Analysis of protein-ligand interactions based on the atomic coordinates (*Laskowski and Swindells, 2011*) showed an increased network of electrostatic and hydrophobic interactions, and three predicted hydrogen bonds, that the inactivated conformation gained with respect to closed conformation (*Figure 4A*). Thus, the nucleotide nestled deeper into the cavity in going from closed to open, and then to inactivated, increasing connectivity between S6C' and CTD and reducing connectivity to the U-motif (*Figure 4B*). Interestingly, ACP in RyR1-ACP/$Ca^{2+}{}_A$ inactivated acquired an interaction with

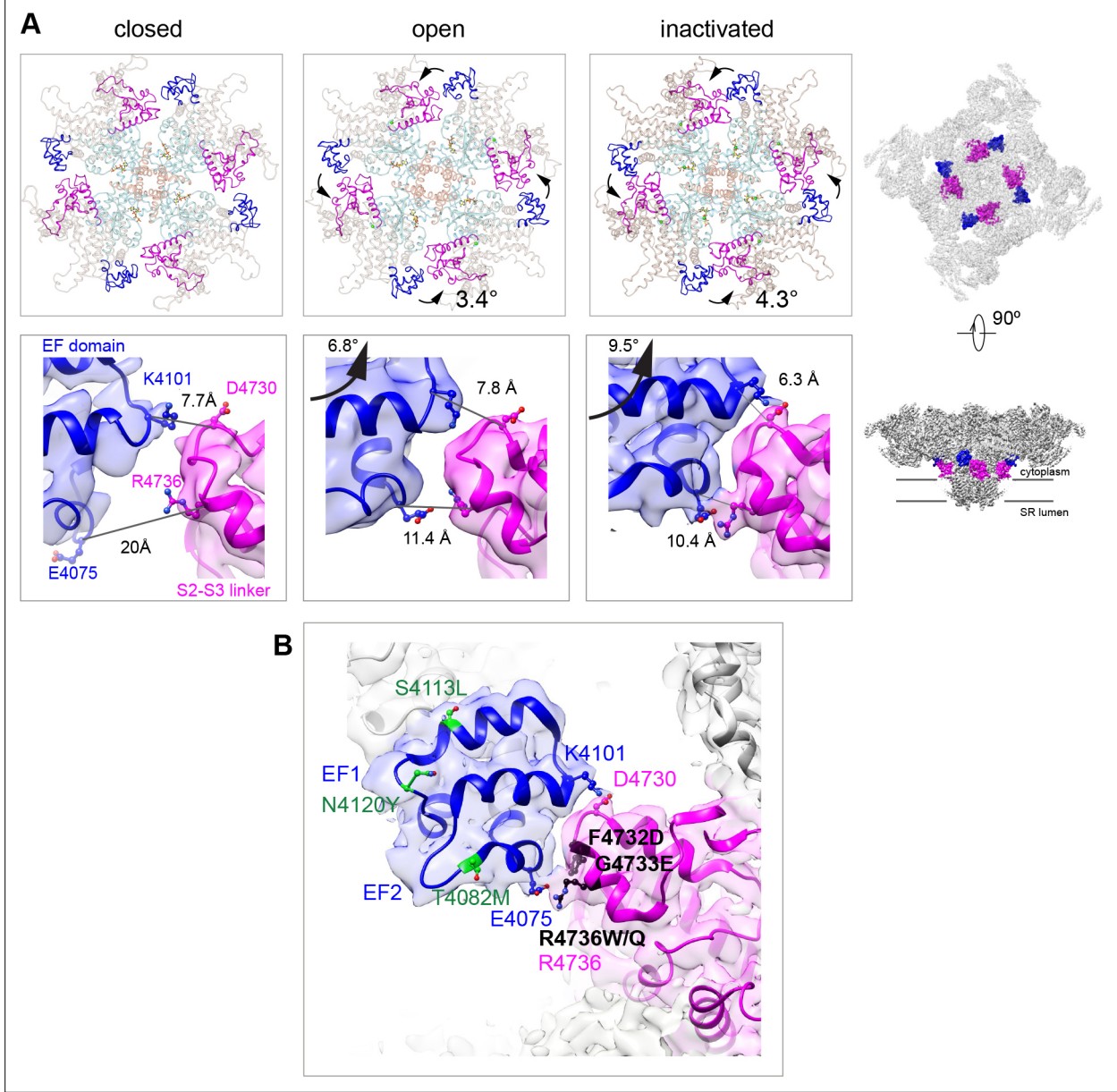

**Figure 3.** Two salt bridges between the EF hands and the S2–S3 loop are determinant for the inactivated state. (**A**) Cytoplasmic sarcoplasmic reticulum (SR) view (top row) and side view (bottom row) highlighting the EF hand domain (blue) and S2–S3 loop (magenta) at the subunit interface. Counterclockwise in-plane and ascending out-of-plane rotations of the central region in the transition from RyR1-ACP/EGTA (closed) to RyR1-ACP/ $Ca^{2+}$ open brings the two domains closer together. Further rotation of the CD (tan color in top row) and its protruding EF hand domain in RyR1-ACP/ $Ca^{2+}$ inactivated brings the two domains in contact. The inter $C\alpha$-$C\alpha$ contact distances illustrate the progressive approximation of the two domains. Two salt bridges, Lys4101-Asp4730 and Glu4075-Arg4736, form in the RyR1-ACP/$Ca^{2+}$-inactivated conformation. The location of the domains in the context of RyR1 is shown on the right panels. (**B**) MH/CCD mutation sites in the interface between the EF hand domain and S2–S3 loop that abolish $Ca^{2+}$-dependent inactivation (black) versus MH/CCD mutations without effect on inactivation (green). The two EF loops are indicated (EF1, EF2). Residues forming the salt bridges are labeled according to domain color. R4736, directly forming the salt bridge, is susceptible to MH mutation.

the backbone carbonyl of His4983, a residue that participates in the C2H2 zinc motif that is central to the CTD (*Figure 4*). Based on the higher map significance of the putative $Ca^{2+}$ ion in our cryo-EM maps obtained at higher $Ca^{2+}$ concentration (*Figure 4—figure supplement 1*) and the enhanced inactivation when ATP is present (*Sitsapesan and Williams, 2000*), the interface between the nucleotide, the CTD, and S6 could be thought of as an atypical low-affinity $Ca^{2+}$ inactivation site. The picture is bound to be more complex as presumably $Mg^{2+}$ present in the cytoplasm would also bind to ATP.

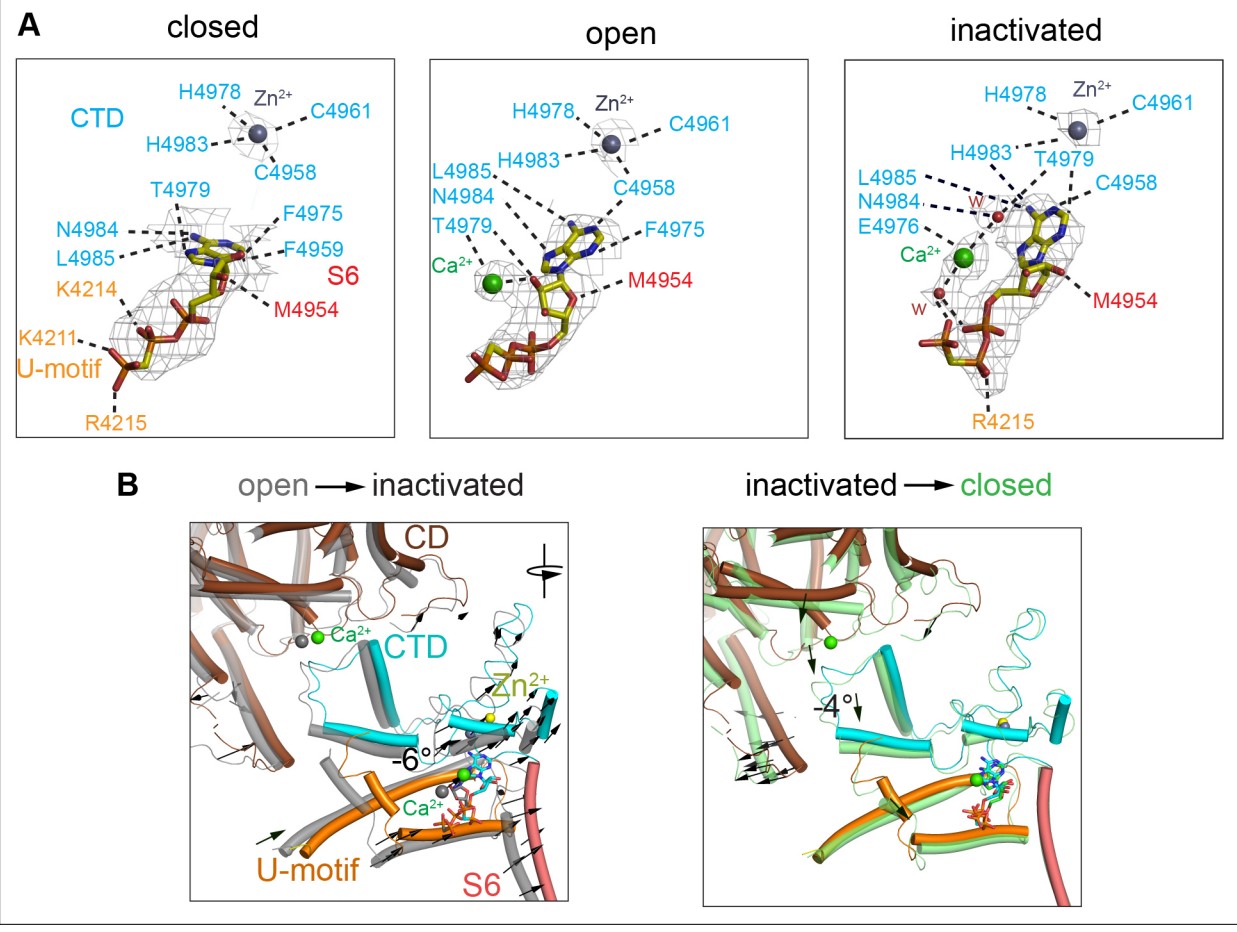

**Figure 4.** Changes in the interaction network of the nucleotide and complexed $Ca^{2+}$. The ATP-binding pocket is formed by the U-motif, CTD, and S6C'. (**A**) cryo-EM densities (gray mesh) of ACP in the different states represented at map significance values (root mean square deviation [RMSD] σ) above 12, 7, and 7, respectively. Residues within 4 Å of either ACP, $Ca^{2+}$, or $Zn^{2+}$ (from the CTD zinc finger) are color-coded according to domain. Notably, an interaction between His4983 (CTD) and ACP is only observed in the inactivated state. Fitted water densities (w) are represented in red. (**B**) Left: ACP goes deeper into the ATP-binding pocket in RyR1-ACP/$Ca^{2+}$ inactivated (colored structure, blue ACP) as compared to the open state (gray structure, red ACP) due to a 6° rotation (see arrows). Right: absence $Ca^{2+}$ in the closed state (green) causes release of the CTD from the CD, resulting in a 4° tilt of the CTD and U-motif (see arrows). Additional reorganization allows closer interaction between ACP and U-motif in the closed state.

The online version of this article includes the following figure supplement(s) for figure 4:

**Figure supplement 1.** Map significance of the ACP and putative $Ca^{2+}$ densities under different conditions.

Thus, under physiological conditions of high local $Ca^{2+}$, competition between the two divalent cations for ATP may take place.

## Protein-lipid interactions within the nanodisc environment

To provide a more physiological environment to the TMD and avoid the presence of detergents, we embedded the protein in scaffold protein MSP1E3D1 that assembled into nanodiscs. The reconstituting lipid was phosphatidylcholine. Refinement focused on the CD-TMD region of RyR1-ACP/$Ca^{2+}{}_A$ inactivated and open, resulting in reconstructions with 3.5 Å and 4.4 Å resolution, respectively, yielded a visible electronfor the nanodisc. Two molecules of the MSP1D1E3 scaffold protein wrapped closely around the TMD in a double-belt arrangement (*Figure 5A*). The larger top belt adopts a quatrefoil shape, with one voltage sensor (S1–S4) in each lobe, while the lower belt is rounder and smaller, following the tapered TMD. The larger footprint of the top half of RyR1's TMD, reflected in the surrounding nanodisc, appears to correlate with the curvature of the membrane around RyR1 observed by electron tomography in its native membrane (*Chen and Kudryashev, 2020*; *Renken et al., 2009*). Even considering the tight fit between the nanodisc and RyR1's TMD, conformational changes were

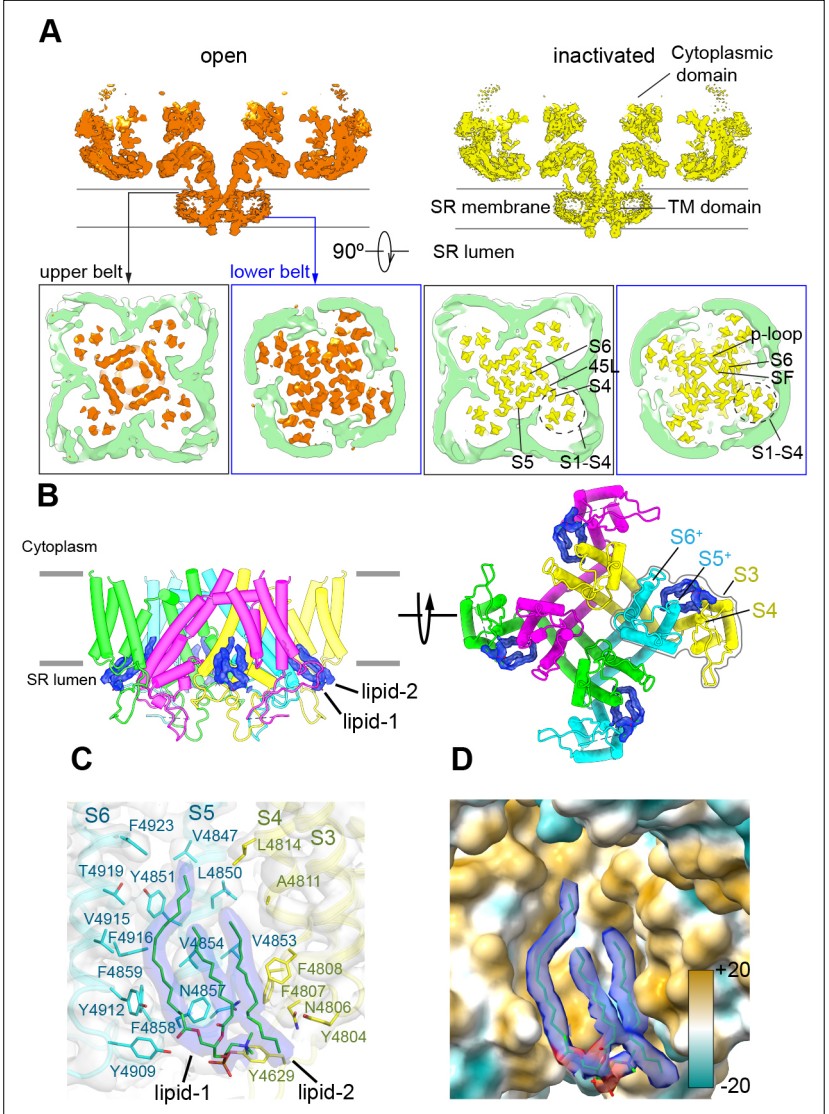

**Figure 5.** The nanodisc environment and visualization of lipids in a crevice of the transmembrane domain (TMD).
(**A**) Top: central slice of the side view of the RyR1-ACP/Ca$^{2+}_A$ open and RyR1-ACP/Ca$^{2+}_A$-inactivated cryogenic
electron microscopy cryo-EM maps highlighting the density corresponding to the upper and lower nanodisc belts.
Bottom: corresponding views seen from the cytoplasmic direction. For clarity, the nanodisc density (green) was
extracted and low-pass filtered to 7 Å resolution. The top belt of the nanodisc expands slightly to accommodate
the conformational change; see also *Figure 5—video 1*. (**B**) Side and luminal views of the TMD of RyR1-ACP/Ca$^{2+}_A$
inactivated with putative lipid densities shown in blue. (**C**) Lipid-binding pocket of RyR1-ACP/Ca$^{2+}_A$ inactivated
lined by lipophilic amino acids from S3 and S4 of the voltage sensor-like domain (S1–S4; yellow) and core helices
S5 and S6 (cyan) from two different protomers. Amino acids within 5 Å from the lipids are shown (sticks) with their
corresponding side chain densities. Densities corresponding to the lipids contoured at 8σ (in blue) are modeled as
a PC (16:0-11:0) for lipid 1 and a 16C acyl chain for lipid 2. (**D**) Molecular lipophilicity potential of the surface lining
the crevice, ranging from hydrophilic (cyan) to hydrophobic (golden). The hydrophobic tails of the lipids are shown
in blue, and the negative electrostatic surface potential of the polar lipid head is shown in red.

The online version of this article includes the following video and figure supplement(s) for figure 5:

**Figure supplement 1.** Lipid-binding pocket lined by the S3/S4 and S5$^+$/S6$^+$ helices in inactivated and closed
structures of RyR suggests a conserved functional site.

**Figure 5—video 1.** Gating-induced conformational changes in nanodisc.

https://elifesciences.org/articles/75568/figures#fig5video1

unhindered and the channel opened within the nanodisc environment. The slight expansion of the top belt of the nanodisc at the level of the ion gate upon opening (*Figure 5—video 1*) reveals a certain degree of plasticity of the scaffold protein.

We resolved two lipid densities at the four inter-subunit interfaces in the RyR1-ACP/Ca$^{2+}$$_A$-inactivated CD-TMD-focused map (*Figure 5B*). The two lipid densities, observed up to 8σ map significance, spanned ~20 Å across the inner leaflet of the SR membrane, at a hydrophobic pocket in the domain-swapped inter-subunit space formed by the S1–S4 bundle and core TMD helices (S5 and S6) (*Figure 5B and C*). One density (lipid 1), encompassing two fatty acyl tail moieties of 16 and 11 carbons with a polar head compatible with a phosphatidylcholine molecule, was resolved near S5/S6 (*Figure 5C*). The lipid likely originates from a longer unsaturated PC (16:0–18:1) used in 50-fold molar excess while embedding RyR1 into nanodiscs. Lipid 2, with a resolved 16-carbon fatty acid tail, was sandwiched between lipid 1 and S3/S4. The lipid's fatty-acyl tails interact extensively with 21 amino acids in the lipophilic pocket formed by S3/S4 and S5/S6 of neighboring subunits, while the density corresponding to the lipid head moiety was positioned within 5 Å from Tyr4629, Asn4857, and Tyr4909 (*Figure 5C*). Together, the lipids covered ~622 Å$^2$ of the 1990 Å$^2$ surface area of the S3/S4-S5/S6 subunit interface forming a hydrophobic crevice (*Figure 5D*). No lipid density was resolved in the RyR1-ACP/Ca$^{2+}$$_A$ open map: besides the lower resolution of this map, its crevice is narrower, as S3 and S4 remodeled the lipid-binding pocket by tilting ~4.4° and ~3.9°, respectively. Furthermore, reorientation in Phe4808 (S4) and Tyr4912 (S6) as modeled for the open state would result in steric clash with lipid 2 and lipid 1, respectively (*Figure 5—figure supplement 1A and B*; lipids superimposed from the RyR1-ACP/Ca$^{2+}$-inactivated reconstruction – shown in *Figure 5—figure supplement 1C* in the same orientation), rearranging or even excluding the lipids in the open channel. A similar observation was reported for the TRPV3 channel, whereby lipids get squeezed out in going from the closed to the open state (*Singh et al., 2018*). We did not resolve lipids within the crevice of RyR1-ACP/EGTA, probably owing to the lower resolution of this reconstruction (4.30 Å), but examination of the crevice indicates no expected hindrance to lipid entry offered by residues Phe4808 and Tyr4912 in the closed state (*Figure 5—figure supplement 1D*). Moreover, we resolved lipids in a higher-resolution (3.27 Å) 3D reconstruction of closed RyR reconstituted in lipids, in this case for the RyR2 isoform (*Figure 5—figure supplement 1E*; PDB ID: 6WOU; *Iyer et al., 2020*). The analogous appearance of the lipid densities in the two isoforms is remarkable (three tubular densities with similar lengths and orientations; compare *Figure 5—figure supplement 1C and E*), suggesting that the interaction of lipids with the RyR TMD crevice is conserved among isoforms, and characteristic of closed and inactivated states.

## Plasticity of transmembrane helices and stability of the luminal mouth of the channel

The TMD helices were examined in the membrane-like environment provided by the nanodisc. Significant perturbations in the α-helical configuration were observed. S4 displays near-3$_{10}$ helix in residues 4807–4813 and (only for the open structure) near-π-helix in residues 4814–4819. S5 displays π-helix in residues 4856–4859. S6 displays π-helix in residues 4921–4928 of S6 (*Figure 6A and B*). In S4, comparison of the three conformations suggests that the degree of over-coiling of the N′ half of S4 is balanced by the uncoiling of its C′ half to near-π-helix in the open structure (*Figure 6A*), which causes the reorientation of Phe4808 into the hydrophobic crevice. In S6, the π-helix region introduces non-canonical i ← i + 5 hydrogen bonding that is energetically less stable than a regular α-helix (*Fodje and Al-Karadaghi, 2002*; *Kumar and Bansal, 2015*; *Figure 6B*), and may facilitate the flexibility of S6 required for its structural transitions.

The three segments departing from α-helical configuration contain Phe clusters (FFF 4807–4809, FF 4858–4859, FFFF 4920–4923) conserved in all RyR isoforms and IP$_3$R intracellular Ca$^{2+}$ release channels; with some of them lining the hydrophobic crevice and forming contacts with the lipids (*Figure 5C*). Deletion of FF 4923–4924 (human sequence; equivalent to rabbit FF 4922–4923) in S6 of RyR1, which is associated to fetal akinesia deformation syndrome or FADS, resulted in loss of Ca$^{2+}$ conductance (*Xu et al., 2020*), which further supports a role for the Phe clusters in the stability of the pore.

The SR luminal loop (4860–4878) between S5 and the P-loop helix (also termed pore helix; residues 4879–4893), proposed to act as a luminal Ca$^{2+}$ sensor (*Sitsapesan and Williams, 1995*) and as an anchorage point for proteins within the SR lumen (*Beard et al., 2009*), contains six acidic residues

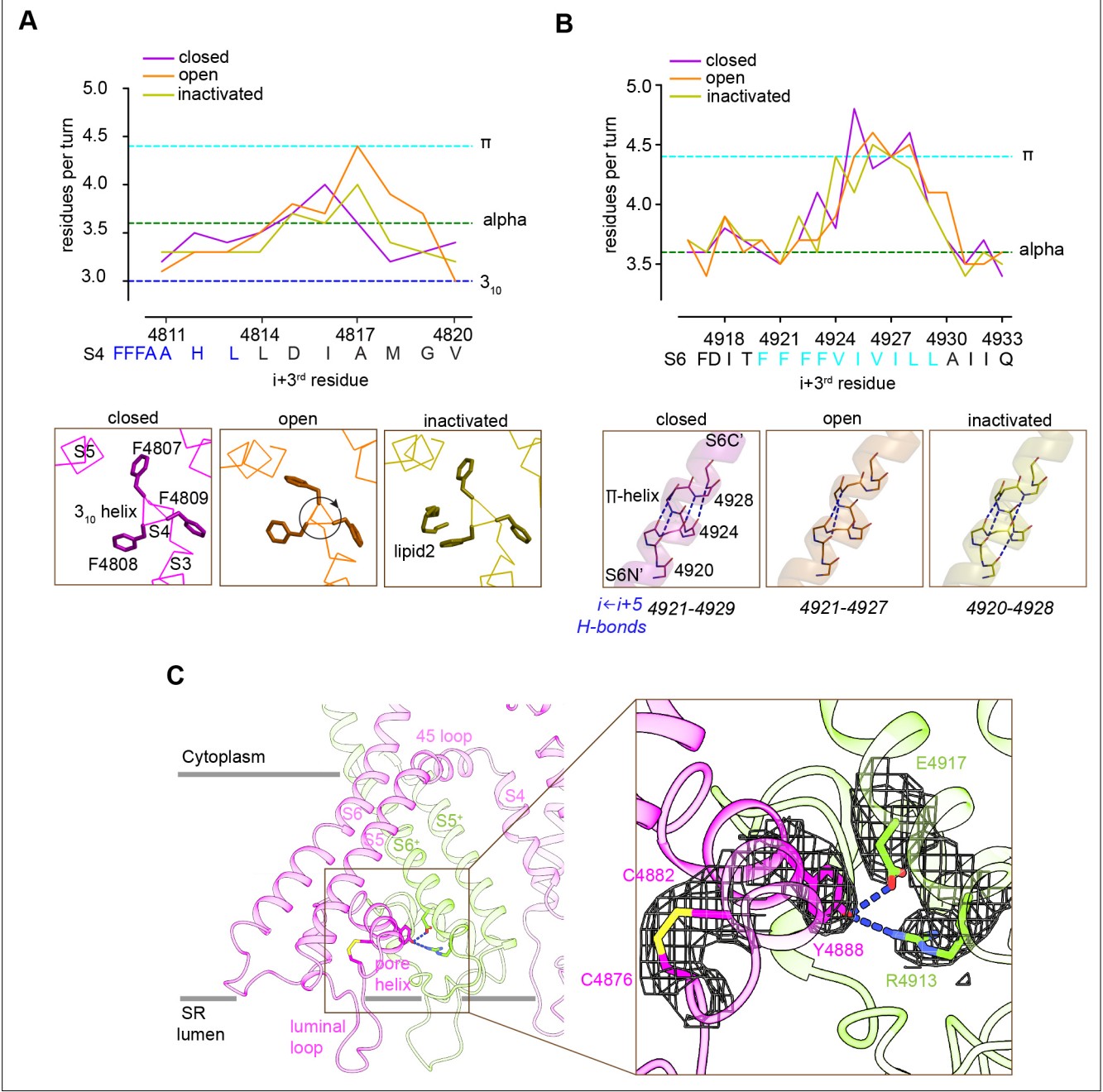

**Figure 6.** Local conformational changes in the transmembrane helices among closed, open, and inactivated conformations. (**A**) Top: residues per turn for S4 reveals a $3_{10}$ helix (4807–4813; blue dashed line) at its N', and helical changes among the three conformations. Residues per turn are computed for a 3-residue moving window. Bottom: a segment of S4 displaying $3_{10}$ helix in FFF (4807–4809). Uncoiling of the S4-C' in the open structure rotates the FFF region, including Phe4808, which interacts with lipid 2 in the inactivated structure. (**B**) Top: the N' region of S6 (residues 4920–4928) contains a tetra Phe motif (4920–4923) followed by a π-helix region. Bottom: the backbone hydrogen-bonding network in the π-helix region in the three distinct conformations; dashed lines show i ← i + 5 hydrogen bonds. (**C**) A Cys4876-Cys4882 disulfide bond formed between the luminal loop (4860–4878) and the pore helix (4879–4893) of the same subunit, and predicted hydrogen bonds between the P-loop (pore helix) and S6 from the neighboring subunit likely contribute to stabilize the luminal mouth of RyR1. The cryo-EM densities are shown in mesh.

(EDEDEPD 4867–4873). This luminal loop is separated from the rest of the TMD, and its acidic residues point to the lumen interior. As our experimental high $Ca^{2+}$ concentrations correspond to these in the SR lumen, this conformation of the luminal loop (*Figure 6C*) is probably a close representation of the native state. The higher-resolution RyR1-ACP/$Ca^{2+}$-inactivated map revealed a disulfide bond (Cys4876-Cys4882) between S5 and the P-loop (*Figure 6C*), and two inter-subunit hydrogen bonds

(Tyr4888-Arg4913 and Tyr4888-Asp4917) between the P-loop and S6. These interactions may stabilize the luminal mouth of the channel, keeping the proper arrangement of the four SR luminal loops, as well as holding S6 through its conformational transitions.

## Discussion

The goal of this work was to understand the mechanism of $Ca^{2+}$-induced inactivation of RyR1 from a structural point of view. Although maximum inhibition occurred at 10 mM $Ca^{2+}$, we selected a concentration closer to the highest $Ca^{2+}$ concentration in physiological compartments, around 1–2 mM free $Ca^{2+}$ (in the SR lumen, cytoplasmic $Ca^{2+}$ nanodomains, and extracellular medium). At 2 mM free $Ca^{2+}$, we discerned two classes within each of the two datasets, with their ion gate either in an open or a closed conformation on account of direct pore analysis. The concentration of $Ca^{2+}$ used, 2 mM, is near the $IC_{50}$ of this cation determined experimentally in the $[^3H]$ryanodine-binding assays in the presence of ACP (the ATP analogue used in our studies), a concentration at which $Ca^{2+}$-activated and $Ca^{2+}$-inactivated RyR1 conformations should coexist. Accordingly, we propose that they correspond with the open-pore and closed-pore conformations observed by cryo-EM, respectively.

Earlier 3D structures of RyR in the open state required synthetic activators such as caffeine (*des Georges et al., 2016*), which by itself suffices to activate the channel (*Xu et al., 2018*), or PCB95 (*Bai et al., 2016*; *Samsó et al., 2009*). Our cryo-EM maps of RyR1 and lipid were obtained in the absence of any nonphysiological activator, and in a nanodisc environment. Given the lipidic environment provided by the nanodisc and the lack of exogenous activators, the structure of RyR1-ACP/$Ca^{2+}$ open (reproduced in two independent datasets, A and B) obtained in the presence of $Ca^{2+}$ and ACP provides an accurate account of the native open state. Overall, our open-state structure validates the previous RyR1 open structures obtained using exogenous activators.

The RyR1-ATP/caffeine/$Ca^{2+}$ dataset, obtained in the presence of 30 µM $Ca^{2+}$, 2 mM ATP, and 5 mM caffeine, also yielded a class with a closed pore, which was termed 'primed' (*des Georges et al., 2016*). An apparently puzzling result is that our structure of RyR1-ACP/$Ca^{2+}$ inactivated is similar to the primed conformation. Both consist of an occupied $Ca^{2+}$-binding site with a closed pore. The flexion angle of the primed structure (e.g., PDB ID: 5TAQ), around –3°, is also more akin to an open than a closed state. In these studies, caffeine was used as a means to attain the open state of the channel as caffeine sensitizes RyR1 to $Ca^{2+}$ and increases opening probability. However, caffeine also affects inactivation and shifts the entire bell-shaped curve of RyR1 activation as a function of $Ca^{2+}$ concentration to lower $Ca^{2+}$ concentrations: from a maximum at 30 µM $Ca^{2+}$, to well below 10 µM $Ca^{2+}$ in the presence of saturating caffeine (*Meissner et al., 1997*). Thus, 30 µM $Ca^{2+}$ in the presence of caffeine is on the descending branch of the curve, which corresponds to partially inactivating conditions. According to this, the RyR1 channel conformation qualified as 'primed' in the earlier study is, very likely, inactivated and appears to correspond to the RyR1-ACP/$Ca^{2+}$-inactivated conformation characterized here.

Activation and inactivation appear to proceed under an integrated mechanism, where inactivation depends on prior activation. Both RyR1 open and inactivated conformations showed that $Ca^{2+}$ bound at the high-affinity site in the CD/CTD interface, which joined these and their proximal domains (EF, U-motif, and S6C') into a more rigid central block. In activation, CD and CTD rotate toward each other closing around the $Ca^{2+}$ ion; the rotation of the CTD pulls S6C' outwards, opening the pore. In inactivation, the central block rotates further into the arc initiated by the CD alone, which now pushes the S6 helices toward the central axis. The movement is similar to pushing down the levers of a winged corkscrew while pushing toward the central rod (see schematics in *Figure 2D*), closing the channel in a distinctive conformation different from the EGTA-closed resting state. This state, with a closed pore and $Ca^{2+}$ bound to the activation site, can no longer be activated by $Ca^{2+}$, which is the hallmark of an inactivated state.

Comparison of the open, closed, and inactivated conformations uncovers other novel features. A strong density connected to ACP under the high $Ca^{2+}$ conditions, which we attribute to hydrated $Ca^{2+}$, progressively supports more interactions with the CTD and S6, increasing cohesiveness of the central block (in the order inactivated >open >closed; *Figure 4*), which could help to explain the more robust $Ca^{2+}$- induced inactivation reported in the presence of ATP (*Sitsapesan and Williams, 2000*). In this context, it is important to keep in mind that $Mg^{2+}$ at physiological concentration also binds to the nucleotide-binding pocket of RyR1, which we can confirm with our 3D reconstruction of RyR1-ACP/$Mg^{2+}$ (PDB ID: 7K0S). While in general ATP has a higher affinity for $Mg^{2+}$ than for $Ca^{2+}$, the $Ca^{2+}$

occupancy of this site in the inactivated state is significant (RMSD 15; *Figure 4—figure supplement 1*). Thus, it is difficult to predict how both cations may compete for the same site, something that will require further study. Besides the high-affinity $Ca^{2+}$-binding site and the nucleotide-binding site, there were no other obvious densities in the resolved regions of the RyR1 structure that could account for a bound $Ca^{2+}$ ion, or clusters of negatively charged residues that could support further $Ca^{2+}$-mediated conformational changes.

$Mg^{2+}$ is a potent inhibitor of RyR1, and a mechanism of $Mg^{2+}$ competition for the $Ca^{2+}$ binding site was proposed (*Lamb and Stephenson, 1991*; *Meissner et al., 1986*). We carried out a 3D reconstruction of RyR1-ACP/$Mg^{2+}$ (PDB ID: 7K0S) and found that, at a physiological concentration of $Mg^{2+}$, the high-affinity $Ca^{2+}$-binding site remains empty. Importantly, the $Mg^{2+}$-inhibited conformation of RyR1 is clearly distinct from the $Ca^{2+}$-inactivated RyR1 and bears more resemblance to the EGTA-closed conformation.

The EF hand domain did not have $Ca^{2+}$ bound in the high $Ca^{2+}$ datasets in keeping with the low affinity of $Ca^{2+}$ to this site (*Xiong et al., 1998*). However, this protruding domain, with its sequence between the CD and the U-motif, appears to play a distinctive role in inactivation that derives from its further anticlockwise and out-of-plane rotations with respect to the open state (*Figure 3*). This conformation brings the EF hand domain in contact with the cytoplasmic loop between the S2 and S3 transmembrane helices of the neighboring subunit, forming two inter-subunit electrostatic interactions (*Figure 3*). One possible scenario is that the energy landscape of the open channel allows for overshoot of the opening motion, allowing interaction of the EF hand and S2–S3 loop domains by such salt bridges. These appear to stabilize the inactivated conformation by providing an extra linkage between subunits, and between the cytoplasmic assembly and TMD. Functional studies of MH mutations in the S2–S3 loop and EF hand domains identified a subset of mutations that altered RyR1 function by impairing $Ca^{2+}$ inactivation (*Gomez et al., 2016*). Interestingly, only this subset of residues, when mutated, would impair the inter-domain interactions that we identified, which supports the proposed role of the two salt bridges (Glu4075-Arg4736 and Lys4101-Asp4730) in stabilizing the inactivated state of RyR1.

In the TMD, owing to the nanodisc environment, two lipids in a crevice between S3/S4 of one subunit and S5/S6 of the neighboring subunit were resolved in the inactivated state. The hydrophobic nature of this crevice suggests that lipids may help to stabilize the TMD in the inactivated state. Although we could not resolve lipids in the closed state probably owing to the lower resolution of this 3D reconstruction, we observed lipids for the RyR2 isoform reconstituted in nanodiscs under closed-state conditions (*Iyer et al., 2020*). The orientation of the two lipids in the two isoforms of RyR is identical. In the open state, the hydrophobic crevice is narrower, and Phe4808 of S4 adopts a different orientation that would clash with lipid 2, suggesting rearrangement in the open state. A similar lipid exclusion of bound lipid in the open channel was reported for the TRPV3 channel (*Singh et al., 2018*). Thus, the closed-state-dependent occupation of the TMD crevice by lipids could constitute a common feature among 6-TMD $Ca^{2+}$ channels.

Departure from α-helix geometry was also present in segments of S4 (near-$3_{10}$ helix and near-π-helix), and S5 and S6 (π-helix). In the case of S4, helical structural transitions correlate with the closed, open, and inactivated conformations. The dynamism of the transmembrane helices through the gating transitions appears to be supported by anchoring of Phe motifs in S4, S5, and S6 to lipids in the membrane. In addition, inter-subunit interactions including disulfide bridges inter-connect the 4860–4878 SR luminal loop, the P-loop, and S6N' around the luminal mouth of the channel where the four protomers converge.

Functional studies employing single channels embedded in lipid bilayers show rapid channel inactivation following an RyR1 $Ca^{2+}$ release event. Therefore, after RyR1 opening, a refractory period is needed to relieve inactivation and recover the ability to activate again (*Laver and Lamb, 1998*; *Ríos et al., 2008*; *Schiefer et al., 1995*; *Sitsapesan and Williams, 2000*). The 3D reconstructions reported here provide a structural basis for this refractoriness: we hypothesize that the 3D reconstructions reported here, combined with the time course of $Ca^{2+}$ release, provide a mechanism for this refractoriness as follows. When RyR1 opens, $Ca^{2+}$ concentration in its surrounding nanodomain increases rapidly, and time from $Ca^{2+}$ release onset also increases. Both augment occupancy of the high-affinity $Ca^{2+}$-binding site and the probability of a full conformational change of the CD/CTD block induced by $Ca^{2+}$, which in turn increases the successful formation of the inter-subunit salt bridges, 'sealing' the

transition of the channel to the inactivated state. At this time, the $Ca^{2+}$-inactivated state is in a distinctive locked closed conformation while the high-affinity $Ca^{2+}$-binding site is still occupied. This renders this closed conformation unable to be activated by $Ca^{2+}$ as long as $Ca^{2+}$ occupies the high-affinity site. In this way, $Ca^{2+}$ permeation through the RyR1 provides negative feedback through the same binding site. $Ca^{2+}$-dependent inactivation is often observed in $Ca^{2+}$ permeation pathways, probably to limit cytosolic $Ca^{2+}$ overload that could be detrimental and life-threatening (*Dick et al., 2016*; *Gomez et al., 2016*). Together with previous investigations (*Gomez et al., 2016*; *Gomez and Yamaguchi, 2014*), we propose a structural mechanism for how naturally occurring mutations disturb RyR1 inactivation producing $Ca^{2+}$ dysregulation and muscle disease. Overall, our study provides a structural basis to understand the transitions from the closed, to the open, and then to the $Ca^{2+}$-inactivated state of the RyR1 at high resolution.

# Materials and methods

## Key resources table

| Reagent type (species) or resource | Designation | Source or reference | Identifiers | Additional information |
|---|---|---|---|---|
| Strain, strain background (*Oryctolagus cuniculus*, mixed gender) | New Zealand White | Charles River | NZW 052 | |
| Recombinant DNA reagent | Membrane scaffold protein plasmid pMSP1E3D1 | Addgene | Cat# 20066 | |
| Chemical compound, drug | Ryanodine, [9,21-$^3$H(N)]-, 250 μCi | PerkinElmer | Part# NET950250UC | |
| Chemical compound, drug | Adenosine-5'-[(α,β)-methyleno]diphosphate, Sodium salt | Jena Bioscience | Cat# NU-420-25 | |
| Chemical compound, drug | 16:0-18:1 PC(POPC) | Avanti Polar Lipids | Cat# 850457 | |
| Chemical compound, drug | 3-[(3-Cholamidopropyl) dimethylammonio]–1-propanesulfonate (CHAPS) | Sigma-Aldrich | Cat# 220201 | |
| Chemical compound, drug | HiTrap Heparin HP | Cytiva | Cat# 17-0406-01 | |
| Chemical compound, drug | L-α-phosphatidylcholine | Sigma-Aldrich | Cat# P3644 | |
| Chemical compound, drug | Sucrose | Sigma-Aldrich | Cat# S9378 | |
| Software, algorithm | Maxchelator | *Bers et al., 2010* | https://somapp.ucdmc.ucdavis.edu/pharmacology/bers/maxchelator/CaMgATPEGTA-TS.htm | |
| Software, algorithm | MotionCor2 | *Zheng et al., 2017* | | |
| Software, algorithm | Gctf | *Zhang, 2016* | | |
| Software, algorithm | RELION-3.0 | *Scheres, 2012* | | |
| Software, algorithm | PHENIX | *Afonine et al., 2018* | http://www.phenix-online.org/ | |
| Software, algorithm | HELANAL | *Bansal et al., 2000* | http://nucleix.mbu.iisc.ernet.in/helanalplus/index.html | |
| Software, algorithm | Coot | *Emsley et al., 2010* | https://www2.mrc-lmb.cam.ac.uk/personal/pemsley/coot/ | |
| Software, algorithm | UCSF Chimera | *Pettersen et al., 2004* | https://www.cgl.ucsf.edu/chimera/ | |
| Software, algorithm | UCSF ChimeraX | *Pettersen et al., 2021* | https://www.cgl.ucsf.edu/chimerax/ | |
| Software, algorithm | PyMOL | *Schrodinger, 2015* | https://pymol.org/2/ | |
| Software, algorithm | HOLE | *Smart et al., 1993* | http://www.holeprogram.org/ | |

*Continued on next page*

*Continued*

| Reagent type (species) or resource | Designation | Source or reference | Identifiers | Additional information |
| --- | --- | --- | --- | --- |
| Software, algorithm | Adobe Creative Cloud | Adobe | https://www.adobe.com/creativecloud/ | |
| Other | UltraAufoil −1.2/1.3 Holey-Gold 300 mesh grids | Quantifoil, Germany | https://www.quantifoil.com/products/ultrafoil | |

## Reagents

All chemicals were purchased from Thermo Fisher or Sigma-Aldrich except where indicated.

## [$^3$H]Ryanodine binding

RyR1 activity was estimated by measuring the extent of bound $^3$[H]ryanodine in microsomes isolated from rabbit skeletal muscle when incubated with free $Ca^{2+}$ alone (10 μM to 2 mM range), or in the presence of 2 mM ATP or 2 mM AMP-PCP (ACP) sodium salts. Concentrations of total $Ca^{2+}$ added to the reaction mixture were estimated in Maxchelator (https://somapp.ucdmc.ucdavis.edu/pharmacology/bers/maxchelator). Preincubated membrane vesicles (~40 μg) were allowed to bind 5 nM [$^3$H] ryanodine (PerkinElmer) in a buffer containing 50 mM MOPS (pH 7.4), 0.15 M KCl, 0.3 mM EGTA, protease inhibitors, and 2 mM DTT for 3 hr at 37°C. Sample aliquots were diluted sevenfold with an ice-cold wash buffer (0.1 M KCl) before placing onto Whatman GF/B filter papers in a vacuum-operated filtration apparatus. The remaining radioactivity in the filter papers after washing three times with the wash buffer was measured by liquid scintillation counting. Nonspecific ryanodine binding was estimated in the presence of 250 μM unlabeled ryanodine (Calbiochem) and subtracted from the total binding. Data represent the mean specific [$^3$H]ryanodine binding from four independent experiments.

## Purification of RyR1 from rabbit skeletal muscle and reconstitution into nanodiscs

Microsomes were purified from rabbit back and hind leg muscles through differential centrifugation as previously described (*Hu et al., 2021*; *Samsó et al., 2009*). 100 mg frozen membranes were thawed and solubilized in buffer A containing 20 mM MOPS pH 7.4, 1 M NaCl, 9.2% (w/v) CHAPS, 2.3% (w/v) phosphatdylcholine (PC; Sigma), 2 mM DTT, and protease inhibitor cocktail for 15 min at 4°C. The solubilized membranes were centrifuged at 100,000 × *g* for 60 min and the pellet was discarded. Supernatant was layered onto 10–20% (w/v) discontinuous sucrose gradients, prepared in buffer B (buffer A plus 0.5% CHAPS and 0.125% PC). The layered sucrose gradient tubes were ultracentrifuged at 120,000 × *g* for ~20 hr at 4°C to allow RyR1 separation. Fractions containing >95% pure RyR1 were pooled and further purified with a HiTrap Heparin HP Agarose column (GE Healthcare) after a fivefold dilution in salt-free buffer and filtration steps. RyR1 was eluted with buffer B containing 0.9 M NaCl, after washing with 20 column volumes of buffer B with 200 mM NaCl. Peak fractions were flash-frozen and stored at −80°C until reconstitution into nanodiscs and cryo-EM. 1.5–2 mg of RyR1 was purified from 100 mg of SR membrane vesicles. RyR1 purity was estimated with 12.5% SDS-PAGE and negative staining with 0.75% uranyl formate. Protein concentration in purified microsomes and RyR1 fractions was measured with Quick Start Bradford Protein Assay (Bio-Rad). The plasmid encoding for MSP1E3D1, pMSP1E3D1, was purchased from Addgene, and recombinant MSP1E3D1 was purified in *Escherichia coli* using the manufacturer's instructions. RyR1-nanodiscs were obtained by mixing purified RyR1, MSP1E3D1, and POPC (Avanti polar lipids) at a 1:2:50 molar ratio. The mixture was incubated for 1 hr 30 min at 4°C before an overnight dialysis in a CHAPS-free buffer (20 mM MOPS pH 7.4, 635 mM KCl, 2 mM DTT), which contained either 1 mM EGTA +1 mM EDTA for the control 'RyR1-ACP/ EGTA' dataset or 3.7 mM $CaCl_2$ for the RyR1-ACP/$Ca^{2+}$ dataset. The dialyzed RyR1-nanodisc preparations were incubated with ACP (sodium salt) for 30 min prior to plunge freezing, at concentrations of 5 mM ACP (control RyR1-ACP/EGTA dataset) or 2 mM ACP (RyR1-ACP/$Ca^{2+}$ datasets). Free $Ca^{2+}$ was estimated with Maxchelator. Integrity of the nanodisc-embedded channels was examined by negative staining.

## Cryo-EM grid preparation and data acquisition

Cryo-EM grids were cleaned with a customized protocol (*Passmore and Russo, 2016*) prior to glow discharge. Aliquots of 1.25–1.5 µl RyR1-nanodisc were applied onto each side of glow-discharged 300 mesh UltraAufoil –1.2/1.3 Holey-Gold (Quantifoil, Germany). The grids were blotted for 1–1.5 s with an ashless Whatman Grade 540 filter paper in a Vitrobot Mark IV (Thermo Fisher Scientific) and rapidly plunged into liquid ethane. Grid quality and RyR1 sample distribution were assessed on a Tecnai F20 (Thermo Fisher Scientific) electron microscope. Data acquisition was carried out in a Titan Krios transmission electron microscope (Thermo Fisher Scientific) operated at 300 kV and counting mode, with a K3 or K2 detector (Gatan) for the ACP/Ca$^{2+}$$_A$ and ACP/Ca$^{2+}$$_B$ datasets, respectively. A Gatan Quantum Energy Filter (GIF) with a slit width of 20 eV was employed. The ACP/EGTA dataset was collected on a K2 detector and a 20 eV GIF. Datasets were collected in automated mode with the program Latitude (Gatan) with a cumulative electron dose of 70 e$^-$/Å$^2$ applied over 50–60 frames. Image acquisition parameters for RyR1-ACP/EGTA, RyR1-ACP/Ca$^{2+}$$_A$, and RyR2-ACP/Ca$^{2+}$$_B$ datasets are summarized in *Table 1*.

## Single-particle image processing

Gain reference normalization, movie frame alignment, dose weighting, and motion correction of the collected movie stacks were carried out with Motioncor2 (*Zheng et al., 2017*). Contrast transfer function parameters were estimated from non-dose-weighted motion-corrected images using Gctf (*Zhang, 2016*). All subsequent image processing operations were carried out using dose-weighted, motion-corrected micrographs in RELION 3.0 (*Scheres, 2012*). The micrographs were low-pass filtered to 20 Å before automated particle picking. 2D class average templates for autopicking were generated by reference-free 2D classification of 1000 manually picked particles. Autopicked particles with ethane and hexagonal ice-contaminated areas were removed by visual inspection. Particle image sub-stacks required for the focused reconstructions residues 3668–5037, encompassing the CD, U-motif, TMD, and CTDs, were generated using a signal subtraction procedure employed in relion_project module (*Bai et al., 2015*). Particle image stacks of quarter sub-volumes of RyR1 corresponding to a single subunit were generated by particle subtraction following a symmetry expansion step with relion_particle_symmetry_expand tool (*Bai et al., 2015*; *Scheres, 2016*). Composite tetrameric maps were generated from the symmetry-expanded monomeric maps with Chimera (*Pettersen et al., 2004*) vop maximum tool. B-factor applied to the reconstructed maps was estimated with relion_postprocess. Analysis of the TMD was carried out on a focused map of the CD-TMD region. Unfiltered half maps obtained from the final 3D-refinement step in RELION 3.0 were further density modified with PHENIX. Resolve (*Terwilliger et al., 2020*). The reported resolutions of the cryo-EM maps are based on FSC 0.143 criterion (*Scheres and Chen, 2012*). Local resolution was estimated with ResMap (*Kucukelbir et al., 2014*). Pixel size calibration of postprocessed maps was carried out using real space correlation metric of UCSF Chimera based on a published RyR1 cryo-EM map (*des Georges et al., 2016*). Pixel size maxima of 1.07, 1.105, and 1.07 Å were obtained in RyR1-ACP/EGTA, RyR1-ACP/Ca$^{2+}$$_A$, and RyR1-ACP/Ca$^{2+}$$_B$ respectively. Image processing schemes of the RyR1-ACP/Ca$^{2+}$$_A$, RyR1-ACP/Ca$^{2+}$$_B$, and RyR1-ACP/EGTA datasets are summarized in *Figure 1—figure supplement 1*, *Figure 1—figure supplement 3*, and *Figure 1—figure supplement 4*, respectively.

## Model building and structure refinement

The cryo-EM-based atomic models of RyR1 (PDB ID: 5tb3 for RyR1-ACP/EGTA, RyR1-ACP/Ca$^{2+}$ inactivated and 5ta3 for RyR1-ACP/Ca$^{2+}$ open) were used as the initial models for model building. The best resolved symmetry-expanded cryo-EM map for a single subunit in inactivated or open conformation was docked within a RyR1 monomer model with Chimera Fit in map tool. Local density fit of the RyR1 sequence was improved over an iterative process of amino acid fitting in Coot (*Emsley et al., 2010*) alternated with real space refinement in PHENIX (*Afonine et al., 2018*). Four copies of the monomers were docked to the whole RyR1 reconstructions. Real space refinement of the tetrameric models was carried out with secondary structure and Ramachandran restraints. Further manual fitting of the CD, TMD, and CTD (3668–5037) of RyR1 was carried out in Coot. Comprehensive model validation was carried out with PHENIX and PDB validation server at https://validate-rcsb-2.wwpdb.org/ and is summarized in *Table 2*. Molecular lipophilicity potential surfaces were

drawn in ChimeraX (*Ghose et al., 1998*; *Laguerre et al., 1997*; *Pettersen et al., 2021*). Figures were generated with PyMOL (*Schrodinger, 2015*) and Chimera programs (*Pettersen et al., 2004*; *Pettersen et al., 2021*).

## Pore radius and helical geometry measurements

Pore radii were measured for the refined atomic model coordinates of the RyR1 pore region (residues 4821–5037) with the HOLE program (*Smart et al., 1993*). Dot surfaces representing the channel ion permeation pathway were generated with HOLE implemented in Coot, which were reformatted to enable visualization in UCSF Chimera. Residues per turn of S4 and S6 transmembrane helices of RyR1 were calculated with HELANAL (*Bansal et al., 2000*).

## Flexion angle measurement

The cytoplasmic shell flexion angles of RyR1 in different conformations were estimated using the procedure indicated in *Steele and Samsó, 2019* with minor changes. The angle was calculated between a diagonal running from the N-terminal domain (residue 348) to the P1 domain (residue 984) from the same subunit, and the horizontal plane.

## Data availability

Tetrameric and focused cryo-EM maps of RyR1-ACP/EGTA, RyR1-ACP/$Ca^{2+}$ inactivated, and RyR1-ACP/$Ca^{2+}$ open have been deposited in the Electron Microscopy Databank (EMDB) with the following accession codes: 22616, 22597 (RyR1-ACP/EGTA tetrameric and focused maps), 25828 (RyR1-ACP/$Ca^{2+}_A$ inactivated; focused map as an additional map), 25830, 25831, 25832 (three subclasses of RyR1-ACP/$Ca^{2+}_A$ inactivated), 25829 (RyR1-ACP/$Ca^{2+}_A$ open), and 25833 (RyR1-ACP/$Ca^{2+}_B$ inactivated). Atomic models generated from the cryo-EM maps have been deposited in the RCSB PDB database with the following accession codes: 7K0T (RyR1-ACP/EGTA), 7TDG (RyR1-ACP/$Ca^{2+}_A$ inactivated), 7TDJ, 7TDI, 7TDK (three subclasses of RyR1-ACP/$Ca^{2+}_A$ inactivated), and 7TDH (RyR1-ACP/$Ca^{2+}_A$ open).

# Acknowledgements

Cryo-grid preparation and screening were carried out at the cryo-EM Unit at Virginia Commonwealth University (VCU) supported by the VCU School of Medicine and MS's funds. Cryo-EM data collection was carried out at the Frederick National Laboratory for Cancer Research supported by contract HSSN261200800001E, at the Molecular Electron Microscopy Core Facility at the University of Virginia (UVA) (supported by NIH U24 GM116790). We thank Drs. Thomas Edwards, Ulrich Baxa, and Adam Wier for cryo-EM data collection at the Frederick National Laboratory. This work was supported by NIH R01 AR068431 and the Muscular Dystrophy Association (MDA 352845) (to MS).

# Additional information

### Funding

| Funder | Grant reference number | Author |
| --- | --- | --- |
| National Institutes of Health | R01 AR068431 | Montserrat Samsó |
| Muscular Dystrophy Association | MDA 352845 | Montserrat Samsó |
| National Institutes of Health | U24 GM116790 | Montserrat Samsó |
| National Institutes of Health | HSSN261200800001E | Montserrat Samsó |

The funders had no role in study design, data collection and interpretation, or the decision to submit the work for publication.

## Author contributions
Ashok R Nayak, Data curation, Formal analysis, Investigation, Validation, Visualization, Writing – original draft, Writing – review and editing; Montserrat Samsó, Conceptualization, Funding acquisition, Investigation, Project administration, Supervision, Validation, Visualization, Writing – original draft, Writing – review and editing

## Author ORCIDs
Ashok R Nayak ⬤ http://orcid.org/0000-0001-9531-5168
Montserrat Samsó ⬤ http://orcid.org/0000-0002-2788-3283

## Ethics
This study was performed in strict accordance with the recommendations in the Guide for the Care and Use of Laboratory Animals of the National Institutes of Health. All of the animals were handled according to approved institutional animal care and use committee (IACUC) protocol #AD10001029 of Virginia Commonwealth. Animals were deeply anesthetized for tissue harvesting, and every effort was made to minimize suffering.

## Decision letter and Author response
Decision letter https://doi.org/10.7554/eLife.75568.sa1
Author response https://doi.org/10.7554/eLife.75568.sa2

# Additional files

## Supplementary files
• Transparent reporting form

## Data availability
The cryo-EM maps and models are available in the EMDB and PDB databases.

The following datasets were generated:

| Author(s) | Year | Dataset title | Dataset URL | Database and Identifier |
|---|---|---|---|---|
| Nayak AR, Samsó M | 2021 | Cryo-EM structure of rabbit RyR1 in the presence of AMP-PCP in nanodisc | https://www.rcsb.org/structure/7K0T | RCSB Protein Data Bank, PDB-7K0T |
| Nayak AR, Samsó M | 2021 | Cryo-EM structure of rabbit RyR1 in the presence of AMP-PCP in nanodisc | https://www.ebi.ac.uk/emdb/EMD-22616 | Electron Microscopy Data Bank, EMD-22616 |
| Nayak AR, Samsó M | 2021 | Focused cryo-EM map of rabbit RyR1 central and transmembrane domains in the presence of AMP-PCP in nanodisc | https://www.ebi.ac.uk/emdb/EMD-22597 | Electron Microscopy Data Bank, EMD-22597 |
| Nayak AR, Samsó M | 2022 | Rabbit RyR1 with AMP-PCP and high Ca2+ embedded in nanodisc in inactivated conformation | https://www.rcsb.org/structure/7TDG | RCSB Protein Data Bank, PDB-7TDG |
| Nayak AR, Samsó M | 2022 | Rabbit RyR1 with AMP-PCP and high Ca2+ embedded in nanodisc in inactivated conformation, (Dataset-A) | https://www.ebi.ac.uk/emdb/EMD-25828 | Electron Microscopy Data Bank, EMD-25828 |
| Nayak AR, Samsó M | 2022 | Rabbit RyR1 with AMP-PCP and high Ca2+ embedded in nanodisc in closed-inactivated conformation class 1 (Dataset-A) | https://www.rcsb.org/structure/7TDJ | RCSB Protein Data Bank, PDB-7TDJ |

*Continued on next page*

*Continued*

| Author(s) | Year | Dataset title | Dataset URL | Database and Identifier |
|---|---|---|---|---|
| Nayak AR, Samsó M | 2022 | Rabbit RyR1 with AMP-PCP and high Ca2+ embedded in nanodisc in closed-inactivated conformation class 1 (Dataset-A) | https://www.ebi.ac.uk/emdb/EMD-25831 | Electron Microscopy Data Bank, EMD-25831 |
| Nayak AR, Samsó M | 2022 | Rabbit RyR1 with AMP-PCP and high Ca2+ embedded in nanodisc in closed-inactivated conformation class 2 (Dataset-A) | https://www.rcsb.org/structure/7TDI | RCSB Protein Data Bank, PDB-7TDI |
| Nayak AR, Samsó M | 2022 | Rabbit RyR1 with AMP-PCP and high Ca2+ embedded in nanodisc in closed-inactivated conformation class 2 (Dataset-A) | https://www.ebi.ac.uk/emdb/EMD-25830 | Electron Microscopy Data Bank, EMD-25830 |
| Nayak AR, Samsó M | 2022 | Rabbit RyR1 with AMP-PCP and high Ca2+ embedded in nanodisc in closed-inactivated conformation class 3 (Dataset-A) | https://www.rcsb.org/structure/7TDK | RCSB Protein Data Bank, PDB-7TDK |
| Nayak AR, Samsó M | 2022 | Rabbit RyR1 with AMP-PCP and high Ca2+ embedded in nanodisc in closed-inactivated conformation class 3 (Dataset-A) | https://www.ebi.ac.uk/emdb/EMD-25832 | Electron Microscopy Data Bank, EMD-25832 |
| Nayak AR, Samsó M | 2022 | Rabbit RyR1 with AMP-PCP and high Ca2+ embedded in nanodisc in open conformation | https://www.rcsb.org/structure/7TDH | RCSB Protein Data Bank, PDB-7TDH |
| Nayak AR, Samsó M | 2022 | Rabbit RyR1 with AMP-PCP and high Ca2+ embedded in nanodisc in open conformation | https://www.ebi.ac.uk/emdb/EMD-25829 | Electron Microscopy Data Bank, EMD-25829 |
| Nayak AR, Samsó M | 2022 | Rabbit RyR1 with AMP-PCP and high Ca2+ embedded in nanodisc in inactivated conformation (Dataset-B) | https://www.ebi.ac.uk/emdb/EMD-25833 | Electron Microscopy Data Bank, EMD-25833 |

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
