## [Editor Report]

This study provides insights into structural changes leading to calcium-dependent inactivation (CDI) in type 1 ryanodine receptors (ryR1). The results nicely rationalize how some disease-causing mutations in RyR1 eliminate CDI of the channel and will be of interest to ion channel structural biologists and physiologists studying skeletal muscle pathologies.

---

## [Decision Letter]

**Decision letter after peer review:**

Thank you for submitting your article "Ca^2+^-inactivation of the mammalian ryanodine receptor type 1 in a lipidic environment revealed by cryo-EM" for consideration by *eLife*. Your article has been reviewed by 2 peer reviewers, one of whom is a member of our Board of Reviewing Editors, and the evaluation has been overseen by Richard Aldrich as the Senior Editor. The reviewers have opted to remain anonymous.

Essential revisions:

1) There is some ambiguity regarding the role of the Ca^2+^ ion in the nucleotide binding pocket in the Ca^2+^-dependent inactivation process. Is this site occupied by Ca^2+^ under physiological conditions or does Mg^2+^ binding prevail under those conditions? Is Ca^2+^ binding to this site necessary for Ca^2+^-dependent inactivation? A control structure under the condition of physiological intracellular Mg^2+^ could provide the information about the Mg^2+^ occupancy at the high-affinity Ca^2+^ binding site and reveal the structural difference between the Ca^2+^-bound state and the Mg^2+^-bound state. The competition between these two types of divalent cations and the difference in their stability, occupancy, and duration in the binding site might underlie this transition. If not possible, an explanation should be provided, and a clearer discussion of the inactivation mechanism is warranted.

2) Provide a rationale/discussions for why lipid does not bind the closed channel and suggestions by Reviewer 2 to improve the quality of information provided in figures.

*Reviewer #1 (Recommendations for the authors):*

1. Lipids were found to bind in the transmembrane domain of RyR1 in the inactivated state, but not the open and closed states. The lack of binding to the open channel was rationalized by a narrower crevice and reorientation of two residues that would result in a steric clash with the lipids. A potential rationale for why lipid does not bind the closed channel is not provided or discussed. For completeness this point should be discussed for the closed RyR1 structure.

2. The role of the Ca^2+^ ion in the nucleotide binding pocket in the Ca^2+^-dependent inactivation process is somewhat vague and ambiguous. Is this site occupied by Ca^2+^ under physiological conditions or does Mg^2+^ binding prevail under those conditions? Is Ca^2+^ binding to this site necessary for Ca^2+^-dependent inactivation? A clearer discussion of this point would be helpful.

*Reviewer #2 (Recommendations for the authors):*

I suggest accepting this manuscript after a revision of the following problems.

1. Since the IC50 value for Ca^2+^ was determined as 1.5 mM in the presence of ACP, why not use higher concentration of Ca^2+^ such as 10 mM to obtain a structure representing a full inactivation state? Please add some explanation.

2. I am curious whether there are any other suspicious densities representing some new low-affinity Ca^2+^ binding sites under 2mM Ca^2+^ condition. Through the new structure the authors excluded that the channel inactivation is due to the reduction of binding affinity of the high affinity site, but they did not provide a new hypothesis to explain how the increase of [Ca^2+^] induces a conformational change from the open state to the inactivation state. In Figure S8, they hinted that the changes of Ca^2+^ occupancy in the high-affinity site and the ATP site are associated with the inactivation. One possibility is that at lower [Ca^2+^] the Ca^2+^ binding is dynamic and transient. Thus, there is not enough time for RyR to make a full transit from the open to the inactivation conformation. More discussion about the inactivation mechanism is needed.

3. Please indicate the positions of I4937, Q4933, and selectivity filter explicitly with arrows in Figure 1b.

4. The coordination of Ca^2+^ shown in Figure 2a is not clear in the current format, especially for the central panel. It is difficult to see the different conformations of Q3970 between the open and the inactivated states. The author can use sticks instead of spheres to present the sidechains involved in the interaction network. Also, label the residues in all three panels.

---

## [Author Response]

Reviewer #1 (Recommendations for the authors):1. Lipids were found to bind in the transmembrane domain of RyR1 in the inactivated state, but not the open and closed states. The lack of binding to the open channel was rationalized by a narrower crevice and reorientation of two residues that would result in a steric clash with the lipids. A potential rationale for why lipid does not bind the closed channel is not provided or discussed. For completeness this point should be discussed for the closed RyR1 structure.

Thank you for enquiring further about this important point. The RyR1-ACP/EGTA had insufficient resolution of to discern lipids, thus we did not mention about these in the closed state. But we have compelling evidence to believe that in fact, lipids bind to the closed channel, based on two reasons: 1/ a higher resolution 3D structure of RyR2 prepared under closed-state conditions shows lipids in an identical configuration, and 2/ close examination of residue’s orientation lining the crevice in RyR1-ACP/EGTA suggests that there should be no steric hindrance for lipid binding in RyR1 in the closed state. We take the identical conformation of lipids in RyR1-inactivated and RyR2-closed as an indication that lipid access characterizes both the closed and inactivated states, but not the open state, and that the binding site is conserved between the RyR1 and RyR2 isoforms. We have expanded on this concept by adding three extra panels to the corresponding supplementary figure (inactivated RyR1, closed RyR1 and closed RyR2 in the same orientation as open RyR1), and added the sentences transcribed below in results and discussion:

Results, Page 7: “We did not resolve lipids within the crevice of RyR1-ACP/EGTA, probably owing to the lower resolution of this reconstruction (4.30 Å), but examination of the crevice indicates no expected hindrance to lipid entry offered by residues Phe4808 and Tyr4912 in the closed state (Figure 5—figure supplement 1 D). Moreover, we resolved lipids in a higher-resolution (3.27 Å) 3D reconstruction of closed RyR reconstituted in lipids, in this case for the RyR2 isoform (Figure 5—figure supplement 1E; PDB ID: 6WOU (Iyer et al., 2020)). The analogous appearance of the lipid densities in the two isoforms is remarkable (three tubular densities with similar lengths and orientations; compare Figure 5—figure supplement 1, panels C and E), suggesting that the interaction of lipids with the RyR TMD crevice is conserved among isoforms, and characteristic of closed and inactivated states.”

Discussion, Page 10: “Although we could not resolve lipids in the closed state probably owing to the lower resolution of this 3D reconstruction, we observed lipids for the RyR2 isoform reconstituted in nanodiscs under closed state conditions (Iyer et al., 2020). The orientation of the two lipids in the two isoforms of RyR is identical.”

We also updated Table 1 to indicate that lipids are present in the closed state.

2. The role of the Ca^2+^ ion in the nucleotide binding pocket in the Ca^2+^-dependent inactivation process is somewhat vague and ambiguous. Is this site occupied by Ca^2+^ under physiological conditions or does Mg^2+^ binding prevail under those conditions? Is Ca^2+^ binding to this site necessary for Ca^2+^-dependent inactivation? A clearer discussion of this point would be helpful.

The interplay between Ca^2+^ and Mg^2+^ is indeed an important question. We carried out a structural determination of RyR1 in the presence of ACP and Mg^2+^. Under physiological concentrations (~1 mM free Mg^2+^), and as one would expect, Mg^2+^ binds to the nucleotide site without producing significant conformational change when compared to RyR1-ACP/EGTA. It is difficult to predict the relative occupancy of this site when both Ca^2+^ and Mg^2+^ are present in mM concentrations; further studies, using techniques other than cryo-EM, will be needed to answer this question.

We should also mention that at physiological concentration of Mg^2+^, the high affinity Ca^2+^ binding site remains empty. Importantly, the Ca^2+^-inactivated and the Mg^2+^-inhibited RyR1s exhibit entirely different conformations. This is an extensive work and out of the scope of the current manuscript; we are preparing a manuscript describing the effect of Mg^2+^. We have added the following paragraphs to address these points:

Discussion, Page 9: “In this context, it is important to keep in mind that Mg^2+^ at physiological concentration also binds to the nucleotide binding pocket of RyR1, which we can confirm with our 3D reconstruction of RyR1-ACP/Mg^2+^ (PDB ID: 7K0S; manuscript in preparation). While in general ATP has higher affinity for Mg^2+^ than for Ca^2+^, the Ca^2+^ occupancy of this site in the inactivated state is significant (RMSD 15; Figure 4—figure supplement 1). Thus, it is difficult to predict out how both cations may compete for the same site, something that will require further study.”

Page 10, discussion: “Mg^2+^ is a potent inhibitor of RyR1, and a mechanism of Mg^2+^ competition for the Ca^2+^ binding site was proposed (Lamb and Stephenson, 1991, Meissner et al., 1986). We carried out a 3D reconstruction of RyR1-ACP/Mg^2+^ (PDB ID: 7K0S; manuscript in preparation) and found that, at a physiological concentration of Mg^2+^, the high affinity Ca^2+^ binding site remains empty. Importantly, the Mg^2+^-inhibited conformation of RyR1 is clearly distinct from the Ca^2+^-inactivated RyR1, and bearing resemblance to the EGTA-closed conformation.”

Reviewer #2 (Recommendations for the authors):I suggest accepting this manuscript after a revision of the following problems.1. Since the IC50 value for Ca^2+^ was determined as 1.5 mM in the presence of ACP, why not use higher concentration of Ca^2+^ such as 10 mM to obtain a structure representing a full inactivation state? Please add some explanation.

We are concerned that 10 mM Ca^2+^ is a concentration that exceeds physiological conditions, and perhaps Ca^2+^ could bind to sites that may not be physiological, or affect other parts of the structure. 1-2 mM Ca^2+^ is the maximum found physiologically (SR store, cytoplasmic Ca^2+^ nanodomains, and extracellular medium). This is a good suggestion for a future experiment in order to test this hypothesis, which should be combined with careful data interpretation.

The Discussion now explains the rationale for selecting 2 mM Ca^2+^:

Discussion, Page 8: “Although maximum inhibition occurred at 10 mM Ca^2+^, we selected a concentration closer to the highest Ca^2+^ concentration in physiological compartments, around 1-2 mM free Ca^2+^ (in the SR lumen, cytoplasmic Ca^2+^ nanodomains, and extracellular medium).”

2. I am curious whether there are any other suspicious densities representing some new low-affinity Ca^2+^ binding sites under 2mM Ca^2+^ condition. Through the new structure the authors excluded that the channel inactivation is due to the reduction of binding affinity of the high affinity site, but they did not provide a new hypothesis to explain how the increase of [Ca^2+^] induces a conformational change from the open state to the inactivation state. In Figure S8, they hinted that the changes of Ca^2+^ occupancy in the high-affinity site and the ATP site are associated with the inactivation. One possibility is that at lower [Ca^2+^] the Ca^2+^ binding is dynamic and transient. Thus, there is not enough time for RyR to make a full transit from the open to the inactivation conformation. More discussion about the inactivation mechanism is needed.

We looked extensively at all the resolved regions of RyR for any extra density surrounded by negative charges and found none. This is mentioned in the manuscript:

Discussion, Page 10: “Besides the high affinity Ca^2+^ binding site and the nucleotide binding site, there were no other obvious densities in the resolved regions of the RyR1 structure that could account for a bound Ca^2+^ ion, or clusters of negatively charged residues that could support further Ca^2+^-mediated conformational changes.”

As the reviewer hints, the temporal dimension seems necessary to place the structural data in context and help to explain inactivation. During the Ca^2+^ transient, Ca^2+^ increases rapidly since the moment when the channel opens, i.e., both Ca^2+^ concentration and time from onset of Ca^2+^ release increase in parallel. Thus, more time since channel opening may increase not only occupancy, but the probability of a full conformational change and successful formation of salt bridges, which would “seal” the transition of the channel to the inactivated conformation.

Page 11: “We hypothesize that the 3D reconstructions reported here, combined with the time course of Ca^2+^ release, provide a mechanism for this refractoriness as follows. When RyR opens, Ca^2+^ concentration in its surrounding nanodomain increases rapidly, and time from Ca^2+^ release onset also increases. Both augment occupancy of the high affinity Ca^2+^ binding site and the probability of a full conformational change of the CD/CTD block induced by Ca^2+^, which in turn increases the successful formation of the inter-subunit salt bridges, “sealing” the transition of the channel to the inactivated state.”

3. Please indicate the positions of I4937, Q4933, and selectivity filter explicitly with arrows in Figure 1b.

Thank you, we have indicated such residues and selectivity filter in the figure.

4. The coordination of Ca^2+^ shown in Figure 2a is not clear in the current format, especially for the central panel. It is difficult to see the different conformations of Q3970 between the open and the inactivated states. The author can use sticks instead of spheres to present the sidechains involved in the interaction network. Also, label the residues in all three panels.

Thank you for the observation and suggestion. Figure 2a has now sticks instead of spheres, and Figure S9 has spheres (i.e., we switched such panels between the main and supplemental figures). We also labeled the relevant residues in all three panels for both main and supplemental figures. Figure legends have been updated accordingly.